# The Evolving Microbiome of Dental Caries

**DOI:** 10.3390/microorganisms12010121

**Published:** 2024-01-07

**Authors:** Grace Spatafora, Yihong Li, Xuesong He, Annie Cowan, Anne C. R. Tanner

**Affiliations:** 1Biology and Program in Molecular Biology and Biochemistry, Middlebury College, Middlebury, VT 05753, USA; 2Department of Public and Ecosystem Health, Cornell University, Ithaca, NY 14853, USA; yihong.li@cornell.edu; 3ADA-Forsyth Institute, Cambridge, MA 02142, USA; xhe@forsyth.org; 4The Mayo Clinic College of Medicine and Science, Rochester, MN 55905, USA

**Keywords:** caries, microbiology, ecology, treatment, *Streptococcus mutans*

## Abstract

Dental caries is a significant oral and public health problem worldwide, especially in low-income populations. The risk of dental caries increases with frequent intake of dietary carbohydrates, including sugars, leading to increased acidity and disruption of the symbiotic diverse and complex microbial community of health. Excess acid production leads to a dysbiotic shift in the bacterial biofilm composition, demineralization of tooth structure, and cavities. Highly acidic and acid-tolerant species associated with caries include *Streptococcus mutans*, *Lactobacillus*, *Actinomyces*, *Bifidobacterium*, and *Scardovia* species. The differences in microbiotas depend on tooth site, extent of carious lesions, and rate of disease progression. Metagenomics and metatranscriptomics not only reveal the structure and genetic potential of the caries-associated microbiome, but, more importantly, capture the genetic makeup of the metabolically active microbiome in lesion sites. Due to its multifactorial nature, caries has been difficult to prevent. The use of topical fluoride has had a significant impact on reducing caries in clinical settings, but the approach is costly; the results are less sustainable for high-caries-risk individuals, especially children. Developing treatment regimens that specifically target *S. mutans* and other acidogenic bacteria, such as using nanoparticles, show promise in altering the cariogenic microbiome, thereby combatting the disease.

## 1. Introduction

Dental caries is a multifactorial disease associated with the structure of deciduous and permanent teeth, the pathogenicity of oral microbial composition, and micro-environments of the oral cavity influenced by sugar, salivary, and genetic factors. While the initial signs of caries affect the outer enamel surfaces of teeth first, in older adults the exposed dentin of root surfaces can be the main risk site for caries initiation [1,2]. Untreated caries can progress into the tooth root canal and produce abscesses. Bacteria produce acids as a byproduct of their metabolism of dietary carbohydrates. Since bacterial metabolism is the source of acid that precedes caries, the systemic host response to bacterial infection plays a significant role in dental caries risk. Host-based factors may be genetic or related to medication side effects, other diseases, or exposure to environmental toxins that can weaken tooth structure. Thus, bacterial composition, diet, and the host defense system are critical factors that can directly or indirectly influence the clinical outcome of the destruction of tooth substance.

Decades of scientific investigations have indicated that sugar consumption is the most significant contributing factor to disease initiation and progression. Unhealthy behaviors, poor oral hygiene, and lack of fluoride exposure are substantial contributing factors to the disease process [2]. The global burden of sugar-related dental caries based on data from 168 countries has provided new evidence that confirms the correlation between the excessive consumption of sugar and dental caries and documents the many dimensions of disparities and financial burdens in dental caries [3].

Furthermore, the global burden of disease suggests that dental caries and untreated caries are responsible for more than 26% of the total dental disease burden and 2.7 million disability-adjusted life years which cost over 172 billion U.S. dollars to treat [4]. On a national level, the U.S. spent over $110 billion in 2016 on dental visits [5], of which over 60% was directed toward treating dental caries. Many high-income countries have a lower disease burden caused by fermentable mono- and disaccharide [MDS]-related dental caries than most middle- and low-income countries [6] and disadvantaged populations and nations with limited access to caries prevention and treatment. Untreated caries can cause pain and infection, impairment of dental function, tooth mortality, missing school hours, and productivity losses. Thus, dental caries remains a serious public health issue with substantial long-term health, economic, and societal impact due to the persistently high prevalence and excessive treatment costs. The disease negatively affects general health and oral health-related quality of life for all age groups [7,8], including infection, impairment of dental function, tooth mortality, missing school hours, and productivity losses. Overall, dental caries remains a serious challenge with substantial long-term health, economic, and societal impact due to its persistently high prevalence and excessive treatment costs [7,8].

In this review, we probe the epidemiology of dental caries and its association with diet. We delve into the history of caries, ecology-based theories about the role of bacteria in caries, and their association with the development of the oral microbiome as a biofilm. We examine the microbiotas associated with enamel and dentin caries and aggressive diseases with the functional activity of bacterial communities. Using an advanced map of the caries microbiome, we explore new control and treatment strategies for modulating disease-associated microbial dysbiosis and the role of therapeutics in alleviating caries and re-establishing a healthy microbiota in the human mouth. The review is based on literature searches in the categories of caries microbiology, epidemiology, etiology, and treatment.

## 2. Background

### 2.1. Epidemiology of Dental Caries

As a bacteria-mediated, sugar-driven, and oral-hygiene-modified dynamic disease, dental caries persists as one of the most common chronic diseases affecting the general population of all age groups. Worldwide, the estimated average prevalence of dental caries of deciduous teeth in 2019 was 43% (38% in high-income countries and 46% in low-income counties) [9]. It is estimated that 514 million children suffer from untreated caries in deciduous teeth worldwide. The estimated age-standardized prevalence of dental caries in permanent teeth was 29.4%, and 2.3 billion adults had untreated caries in permanent teeth [4,10,11].

There has been a decrease in caries in younger age groups in selected regions over the last few decades due to community water fluoridation [12,13,14,15], the widespread availability of fluoride toothpaste [16], reduced sugar intake [17], and increased access to oral health services [18]. In the United States, caries rates slightly decreased in the primary teeth of 2- to 5-year-old children but not in the permanent teeth of 6- to 11-year-old adolescents between 1988 and 2016 (Figure 1A) [19,20,21]. These downward trends, however, were not observed in many low- or middle-income nations when associated with increased sugar consumption during that time period [22].

There were no significant changes in overall caries status in adults aged 20 to 64 years, but a notable increase in older American adults over 65 years [19,20,21] (Figure 1B). Worldwide, billions of people with untreated caries have significantly and dramatically increased the burden on healthcare systems [23]. One contributing factor is an increase in life expectancy, leading to more older people maintaining their teeth, which has an associated caries burden [19,20,21,24,25]. This challenges the healthcare system everywhere, particularly in low- and middle-income countries.

The National Health and Nutrition Examination Survey 2015–2016 data reported that the prevalence of dental caries in permanent teeth increased with age from 9.5% to 96% (Figure 1C) [9]. Significant socioeconomic inequalities exist in caries experience and untreated caries in children and adults. Hispanic youth and adults experienced the highest prevalence of total caries (32.9% at 2–5 years old, 72.8% at 6–8 years old, 69.7% at 12–19 years old, and 86.6% at 20–64 years old) compared with other ethnic groups [9] (Figure 1D). The prevalence was higher among poor and near-poor children and adults than in non-poor counterparts (Figure 1E). Among millions of American children and adults, 16.4% of 6- to 8-year-old children and more than 25% of 20- to 74-year-old adults had at least one untreated decayed tooth (Figure 1F). Non-Hispanic black populations had the highest prevalence of untreated caries in primary teeth (22.4%) and permanent teeth (40.2%) (Figure 1G). The prevalence increased for both total caries and untreated caries as family income levels decreased [9,19,26,27,28] (Figure 1H).

For a long time, the caries diagnosis and risk assessment methods used in epidemiologic studies varied worldwide, resulting in different outcomes for reporting, evaluating, and monitoring disease progression. Many government-funded water fluoridation programs or caries risk assessment programs measured caries management using a risk assessment (CAMBRA^®^ v.4.0) model (cda.org/CAMBRA4) [29] or the Cariogram software program, v.1.0 [30]. Meanwhile caries diagnosis and measurement for population-level studies relied on the DMFT/DMFS (decayed, missing due to caries, and filled tooth/surface) Index [31]. This methodology has been challenged for its lack of sensitivity and discriminatory validity, resulting in substantial underestimation or biased estimation of caries rates and severity. As such, it has had a limited impact on the evaluation of caries experience for high-risk populations [32,33]. In response to these concerns, the International Caries Detection and Assessment System (ICDAS) was developed by an interdisciplinary team of caries researchers, epidemiologists, and clinicians [34,35]. This integrated system has been tested and proved to be easy to use with clearly defined clinical visual criteria for caries detection, a seven-point ordinal scale ranging from sound to extensive cavitation to measure caries severity, and codes to monitor caries activity over time [36]. Along with simplified merged-code options, ICDAS has been increasingly adopted by many countries for caries surveillance and caries risk assessment and management.

In the future, new scientific-evidence-based approaches, such as using new biomarkers for caries pathogens or disease activity in caries epidemiological studies, can evaluate disease incidence and progression and support decision-making at both the individual and public health levels.

### 2.2. Diet and Dental Caries

While it is generally appreciated that the consumption of sugar-based foods and drinks is “bad for your teeth”, the history of dental caries and its relationship with diet goes back to the origins of agriculture. As hunter-gatherers of the European Mesolithic age (9600–4000 BC), humans had a low prevalence of caries impacting ~0–2% of the population [37]. In post-agricultural societies, however, computer modeling analysis indicates that caries incidence has increased to 5–50%, in direct correlation with a concomitant increase in dietary carbohydrate consumption [37] and the availability of processed foods in the Western marketplace in the mid-20th century [38]. This historical sequence of dental caries emergence has been recapitulated in populations under dietary stress. During World War II (1939–45), a decrease in caries was documented in European countries (Czechoslovakia, Denmark, England, Finland, Germany, Holland, Norway, Scotland, and Sweden) where there were reductions in refined carbohydrates including sugar and calorie-restricted diets [39,40]. When the period of diet deprivation ended in the post-war era, caries increased to pre-war levels. The late 1940s post-war caries increase was so alarming that it led to a study to better understand the link between carbohydrates, including sugars, and their frequency of intake over time with caries development in Sweden. This “Vipeholm” study conclusively demonstrated the role of frequency of carbohydrate intake in caries susceptibility [41].

More recently, populations with rapid “Westernization” showed a sharp increase in dental caries experience directly related to changes from eating primarily meat-based diets to those with increased proportions of carbohydrates, as in American Indian and First Nation populations in the US and Canada [42]. Caries-associated carbohydrates go beyond desserts and sugary drinks to include beverages and foods like beer, potato chips, and sweetened yogurts. Individuals who subsist on a Western diet, regardless of their consumption of standard sugar-based foods such as dessert items, are significantly more likely to score higher on the standard decayed, missing, and filled teeth (DMFT) index than those who do not [43].

### 2.3. Plaque pH and Host Factors

In 1944, Stephan suggested that the link between diet and caries was the development of a low plaque pH [44]. His experiments included measuring the pH of plaque over time following a glucose rinse in human volunteers. After an initial drop in pH following the rinse, the acid pH readings gradually returned to baseline levels and plots of the change in pH readings became known as a “Stephan curve”. Of particular interest was the lowering of plaque pH to below 5.5, as this was considered the level of acidity needed for enamel demineralization. Marked differences in plaque acid responses were observed based on caries clinical status including caries-free, slight caries, obvious lesions, and aggressive, advanced lesions. In caries-free subjects, post-glucose rinse acidity remained above pH 5.5, suggesting no enamel demineralization. By contrast, in aggressive caries, the initial resting plaque acidity was below that needed for demineralization, with further lowering after the glucose challenge to under a pH of 4.

But how do sugars, particularly sucrose and fructose, impact the rate of caries development? As a metabolite for caries-associated pathogens, sucrose is easily converted into its monosaccharide constituents, glucose and fructose, each of which can serve as substrates for bacterial fermentation to produce lactic acid byproducts [45]. The combination of acid-producing bacteria and fermentable carbohydrates, some of which can be stored as intracellular polysaccharide, fosters continuous exposure of the host dentition to organic acids for extended periods, including non-meal times [46]. Thus, to understand the impact of the cariogenic process on caries development and treatment, we must understand these and other players that contribute to the caries phenotype.

Host-based caries risk factors, including genetics, stress, and access to fluoridated water, have evolved over time and/or remain disparate for individuals of different socio-economic strata [32]. Populations that cannot afford or have limited access to healthy foods are at higher risk for dental caries. Increasing disease incidence among the socioeconomically disadvantaged derives primarily from host genetics and individual dietary intake [47]. While genetic host-based risk factors can impact the expression of salivary proteins in the tooth pellicle [48,49], dietary risk factors can be related to medications, exposure to environmental toxins, or systemic disease that can weaken tooth structure and resistance to bacterial acid attack. The mechanisms that underlie these risk factors are frequently related to the ability of bacteria to attach to teeth or the resistance of teeth to acidity and demineralization. Bacterial attachment to teeth via the pellicle is a critical step in caries formation, as it encourages subsequent biofilm development. This attachment can be greatly facilitated on the rough surfaces of poorly formed enamel, for example in individuals with genetically determined amelogenesis imperfecta and other conditions leading to tooth malformations.

## 3. Microbiome of Dental Caries

The oral microbiome is at the entryway to the gastrointestinal tract [50]. The evolution of the human body is coincident with, and dependent on, the evolution of its accompanying microbiota [51]. The most important and persistently variable relationships between host and microbe occur within the mouth, esophagus, stomach, small and large intestine, and anus of the gastrointestinal (GI) tract. Every part of this tubular structure, which collectively comprises “the gut”, serves as home to a coalition of microbial species that can co-inhabit its many inherent niches. Research on the microbiomes of the GI tract has focused on the gastric microbiota and the microbiota of the ileum and proximal colon where a dysbiotic microbiome may contribute to autoimmune disorders such as inflammatory bowel disease (IBD) and GI cancers [52,53]. The oral cavity, as the entryway to the digestive tract, plays a particularly crucial role in mediating overall human health, in large part owing to its exposure to diet and stressors in the external environment.

While good oral hygiene marks the cornerstone of a healthy mouth and, for those fortunate enough, a beautiful smile, caries prevention extends well beyond routine practices such as brushing and flossing. The delicate balance of bacterial species interactions in the human oral cavity can become dysbiotic under conditions of oxidative stress, acidic pH, and/or fluctuations in nutrient availability, especially when combined with poor dental hygiene practices and unfavorable host genetics. The oral microbiome, which includes microorganisms on the tongue, palate, and dentition, comprises about 700 different species of bacteria, many of which have not yet been cultured in the laboratory [51]. Ultimately, the ability to study and characterize the healthy oral microbiome is essential for understanding, diagnosing, and treating diseases of the human oral cavity, including dental caries.

Since the first publication of a human metagenomic study in 2006 [54], the field of human microbiome research has exploded. Microbiome research in the 21st century has incorporated research into the dynamic relationship between the oral microbiome and the host environment in health and disease. The total number of manuscripts published in the field of oral health versus the caries-associated microbiome by international researchers has increased markedly in recent years. A substantial “core model” comprised of approximately 60% of the oral taxa has been suggested to contribute to the deleterious shift in bacterial diversity to acidic (cariogenic) from neutral (healthy) pH environments [55].

### 3.1. History of Caries Microbiology

Our understanding of the caries microbiome builds on over 100 years of observations and research studies. The late 1800s and early 1900s marked the “Golden Age of Microbiology” based on the germ theory of disease as enunciated by Louis Pasteur and demonstrated by Robert Koch. Germ theory was expanded to caries by GV Black [56] based on bacterial acid production that could demineralize teeth. A chemo-parasitic theory was proposed by Willoughby Dayton Miller in 1890, based on his experiments that involved soaking teeth in saliva, as a source of bacteria, and bread, as a food source, and observing tooth demineralization [57]. Miller determined that the demineralization he observed came from lactic acid that was produced by bacteria that we now recognize as *Lactobacillus* and *Bifidobacterium*. These bacteria were later cultured from caries by Percy Howe, the first research director at the Forsyth Dental Infirmary for Children founded in 1910 [58]. Howe repeated Miller’s experiments (soaking teeth in saliva with bread) but did not consider the demineralization he observed to be caries based on his observations of children in the clinic with advanced caries. Further, Howe was unable to produce caries in experimental animals using plaque from carious lesions, which led him to question the role of individual species in dental caries. In the 1930s, Theodore Rosebury, “the grandfather of oral microbiology”, reported on the acidogenic properties and acid tolerance of *Lactobacillus acidophilus* of oral and gastrointestinal tract origin [59]. Rosebury, like Howe, did not advocate for specific species in dental infections. Thus, these early investigators recognized the role of bacterial acid production in caries without implicating individual species as had been the Pasteur and Koch tradition.

In the 1920s, Kilian Clark in the UK was interested in whether the same *Bacillus (Lactobacillus) acidophilus* bacteria that Howe had cultured from advanced lesions could initiate caries. Clarke isolated a streptococcus from many initial carious lesions that was capable of demineralizing tooth sections [60]. This new streptococcus was a strong acid producer and was very adherent to the tooth sections so it could resist being washed away by saliva, characteristics that he reported were important for caries pathogens. Clarke named this new species *Streptococcus mutans*, and he considered it capable of causing caries [60]. In studies that involved isolating *S. mutans* during the early stages of caries on enamel surfaces, Clarke was able to isolate lactobacilli from deeper, more advanced lesions, suggesting that different bacteria were involved in the progression of cavities. Clarke’s paper was published in a British journal and, before the era of easy access to international publications, it seems his work was overlooked for several decades.

The essential role of bacteria in caries was demonstrated in the 1940s to 1960s in studies using experimental animals. The association of *Streptococcus mutans* with caries was observed in a series of studies performed mainly at the National Institutes of Health (NIH) in Bethesda MD, USA. The dental wing of the NIH, later the National Institute of Dental Research, then the National Institute of Dental and Craniofacial Research, had its origins in dental caries and was founded in large part to understand the reason why many recruits for World War II in Europe had many carious teeth and too few teeth to serve in the war. The critical importance of bacteria as compared with diet had been demonstrated by the finding of a complete absence of caries in germ-free animals despite maintaining them on a highly cariogenic high-sucrose diet [61]. Paul Keyes then expanded the importance of bacteria in caries in a series of studies using Syrian hamsters and other animals. Keyes summarized many of his experiments in a 1960s publication [62], where he reported that caries could be transferred from animals with caries to those with no lesions. Keyes also noted that, in young weanlings, the source of the caries bacteria was transmitted from the mother. Furthermore, Keyes observed that taking plaque from caries in one animal could induce caries in non-diseased animals, and that antibiotic administration inhibited lesions from developing. Caries was also suppressed in pups from mothers with lower caries experience [62]. In collaboration with Fitzgerald, Keyes reported that the bacterium responsible for experimental caries was an acidogenic streptococcus [63], which Keyes observed could by itself induce caries in germ-free animals. It was not until the late 1960s that Edwardsson made the discovery that the *S. mutans* he isolated from humans was the same species that Keyes described from carious lesions in experimental animals, and also the same species that was described by Clarke in 1924 [64]. Numerous clinical studies have demonstrated the strong association between *S. mutans* and dental caries [65,66,67,68]. Hence, the concept of a key role for *S. mutans* in dental caries was established.

The development of a caries-associated microbiome was subsequently described using an ecological plaque hypothesis [69]. This concept was based, in part, on chemostat studies and the growth of a mixture of nine oral bacteria pulsed with glucose at different pHs over time [70]. The experimental findings indicated that when a healthy resting oral pH of 7.0 was maintained, *Streptococcus gordonii*, which is acidogenic but not very acid-tolerant, dominated the community. When the acidity of the medium was not controlled in the experiment, however, the pH dropped to 4.5 and the acid-tolerant species *S. mutans*, *Lactobacillus casei*, and *Veillonella dispar* dominated the community [70]. According to this model, the metabolism of all acid-producing species in the nine-species cocktail lowered the environmental pH, but only the acid-tolerant species could survive. Thus, the oral biofilm under acid stress undergoes a selective succession that favors acid-tolerant species. In accordance with the ecological plaque hypothesis, it is dietary carbohydrates, particularly sucrose, that are the drivers of acid production in tooth-associated microbial communities. Consistent with this hypothesis are the noticeable shifts in biofilm architecture and microbial population dynamics that were observed after the consumption of a meal [71].

### 3.2. Development of the Tooth-Associated Microbiome Leading to Caries

The tooth-associated microbiome develops in stages that ultimately either defines health (caries-free) or disease (caries-associated). An association between clinical outcomes and microbial changes in oral biofilms under dietary stress was described in 2011 as an extension of the caries ecological hypothesis as proposed by Takahashi and Nyvad [72,73]. The authors described the earlier stages of microbiome change as reversible, depending on the availability and duration of dietary carbohydrates in the mouth.

#### 3.2.1. Pioneer Species

The dynamic stability stage of the extended ecological plaque hypothesis represents health, as dietary carbohydrate challenges are counterbalanced by microbial community activity to rebalance any acid produced. This stage encompasses colonization of the dentition by pioneer species in the tooth pellicle. As described below, microorganisms in the pellicle are comprised mainly of non-mutans streptococci and *Actinomyces* species.

Bacteria can be detected on teeth within two hours after cleaning, and colonization is initiated after the formation of a dental pellicle, which is induced by healthy host salivary secretions [74,75]. As the only non-shedding surface in the human body, the teeth are protected by a hard outer coating of hydroxyapatite, or calcium phosphate, which resists force and wear that is inherent to chewing and swallowing. Included in the pellicle are proteins, often negatively charged glycoproteins, that are produced in the saliva and settle on teeth as biopolymers [75]. The resulting saliva-based film is deposited immediately on the enamel surface shortly after tooth eruption, within minutes after routine brushing, and reaches maturation within 120 min [76]. The tooth pellicle is believed to have evolved to protect the tooth surface from acids produced by local microorganisms [69]; yet, microbes were able to exploit this host defense mechanism by using the pellicle as a substrate for attachment, thereby enhancing the bacterial colonization of tooth surfaces. The host-derived pellicle, which is both diet-induced and individual-specific, serves as the first stage in the development of the plaque biofilm.

The next stage involves colonization of the tooth by so-called “pioneer” species. Most pioneer species have evolved cell-surface ligands which interact directly with proteins as receptors inherent in the dental pellicle. These receptors, deemed “cryptitopes”, often do not appear until after proteins in the salivary pellicle have become absorbed on the hydroxyapatite surface of the tooth, thereby changing protein conformation to foster subsequent bacterial adhesion [77,78]. The attachment process begins with reversible adhesion of bacteria to the tooth via physio-chemical interactions, such as hydrogen bonds, hydrophobic interactions, calcium bridges, and van der Waals forces, that collectively strengthen ligand-receptor-mediated surface attachments.

Pioneer species whose presence is consistent with caries-free sites and gingival health include *Actinomyces naeslundii*, *Actinomyces oris*, *Streptococcus gordonii*, *Streptococcus sanguinis*, and *Streptococcus oralis* [79,80] (Figure 2). These species undergo adhesin-receptor-mediated associations with proteins in the tooth pellicle, thus forming the initial biofilm layer, which, in turn, can provide specific bindings sites for subsequent bacterial colonization mediated by stronger physical interactions between coaggregation receptor polysaccharides (RPS) of the group A streptococci, the oral streptococci, and the fimbriae of most *Actinomyces* species [81].

Pioneer streptococci species can dominate the supragingival regions of teeth and appear to be uniquely attuned to the fluctuating conditions of the oral cavity due to their advanced ability to regulate gene expression and protein production for attachment to the dental pellicle [82]. For example, the regulatory mechanisms of *S. gordonii* allow it to thrive under a certain set of environmental conditions [83]. Similar mechanisms in other species help define which bacteria can co-inhabit the same oral niche [84]. Of the pioneer species, *S. gordonii* is persistently adherent to the tooth enamel, given its high affinity for proteins in the tooth pellicle. *S. gordonii* can utilize adhesion proteins (i.e., Has) and amylase-binding protein A (AbpA) to mediate receptor–ligand attachments to teeth. The Has adhesion receptors of *S. gordonii* can bind specifically to the terminal sialic acid of host sialoglycoconjugates [80], whereas AbpA interacts directly with human α-amylase, a salivary enzyme that is paramount for food breakdown [85]. Due to the covalent nature of these interactions, *S. gordonii* was found to firmly attach to the tooth, comprising up to 70% of the bacterial community in this early dynamic stability stage of plaque development [85].

#### 3.2.2. Early Colonizers

An important step in biofilm development involves a substantial increase in the quantity and variety of streptococci in the dental microbiome [86]. Bacterial adhesion to proteins in the pellicle extends beyond the more typical ligand–receptor interactions and includes cooperative cross-species relationships like that between *S. sanguinis* and *S. oralis*, other pioneer colonizers of the caries biofilm. *S. gordonii*, *S. sanguinis*, and *S. oralis* are facultative anerobes, which is crucial to their pioneering status since they can initiate biofilm formation on the exposed tooth surface and survive oxygen deprivation as the biofilm develops and matures. Furthermore, *S. sanguinis* and *S. oralis* are considered commensal species of the healthy oral microbiome, and they maintain genetic factors which sustain their attachment to the dentition, especially by expressing a variety of Abps [85,87]. More importantly, *S. gordonii*, *S. sanguinis*, and *S. oralis* can co-aggregate and form an intricate system for quorum sensing as the plaque biofilm develops. The *comCDE* operon, which is highly conserved across the streptococci, encodes an essential competence-stimulating peptide (CSP) in this trio. The peptide is a product of the *comC* gene, which increases the cell division rate and regulates a primary set of quorum-sensing genes [79]. This allows these streptococci to interact directly with one another, so they may become sensitive to the environmental conditions around them and, most importantly, regulate the pH of the oral environment. Major mechanisms for buffering local acidity include the production of ammonia by arginine deaminase from *Streptococcus sanguinis* and by urease from *Actinomyces naeslundii* [88,89,90]. Thus, the interspecies communication among pioneer species is what allows them to survive in the developing biofilm, in large part because of their abilities to regulate and neutralize local acid production.

*Veillonella tobetsuensis*, a relatively new species closely related to *V. dispar* and *Veillonella parvula* [91] and likely previously recognized as one of these, is another early colonizer of human dentition. Further, *Veillonella* species, including *V. tobetsuensis*, become especially well established in active carious lesions by selectively allowing species growth in response to the increasingly acidic environment. The cellular machinery for *Veillonella* growth and division can be induced by association with *Streptococcus* species within the tooth pellicle [92]. Hence, biofilm architecture, which is dynamic during the later stages of caries progression, can have a direct impact on the composition and survival of individual species on the tooth surface, thereby shaping the landscape of the plaque microbiota to favor either health or disease depending on the rate of environmental acidification.

#### 3.2.3. Early Colonizers and the Acidogenic Stage of Plaque Biofilm Development

While pioneer species are the first to colonize the tooth pellicle, the appearance of early colonizer microorganisms in succession parallels changes in the local plaque environment that derive from the amassing pioneer species; such accumulation generates conditions no longer favorable for survival of the pioneer species owing to oxygen and nutrient deprivation as well as increased host sugar consumption [69]. The initial horizontal coaggregation of bacteria over the tooth surface is followed by vertical coaggregation away from the tooth surface [93], resulting in biofilm build up that selects for persistent metabolically diverse species that favor growth at low pH. Co-adhesion between the co-aggregating pioneer and early colonizers often occurs by uni- or bimodal protein lectin–oligosaccharide receptor interactions, which allow for the development of a structured biofilm architecture [94].

*S. gordonii*, *S. sanguinis*, and *S. oralis*, as non-mutans streptococci, are often considered “accessory pathogens” because of their contributions to plaque architecture. Importantly, their interactions are often species-specific, such that the attachment of an early colonizer is entirely dependent on the preceding establishment of its partner pioneer colonizer. Thus, the development of the caries biofilm is not so much a one-way, stepwise progression as it is an amalgamation that is dependent on species-specific population dynamics that essentially generate niches for the next stage of colonizers.

The early colonizers produce acidic byproducts from the fermentative metabolism of dietary carbohydrates. A longer acidic challenge leads to suppression of more acid-sensitive oral bacteria and an increase in acid-tolerant species. Biofilm acidity is associated with demineralization of the enamel surface and weakening of the outer protective enamel layer of the tooth. This leads to surface roughening, which can instigate the colonization of more aggressive microbial pathogens [72,95]. This is the stage in which there can be an acid rebalance of the bacterial community so that enamel remineralization and any clinical signs of caries are reversed.

#### 3.2.4. Secondary Colonizers and the Aciduric Stage of Plaque Biofilm Development

Next in the microbial succession are the secondary colonizers, which shift the mainly streptococci, actinomyces, and veillonella microbiota to a more diverse constituency of microorganisms capable of broader metabolism. This stage represents a more acidogenic stage of the ecological plaque hypothesis [72,96]. Together, the secondary colonizers *S. mutans* and *Lactobacillus*, *Bifidobacterium*, and *Scardovia* species contribute to climax communities that define the environmental conditions of the oral cavity, and which can enhance the pathogenic potential of the colonizers involved. If the enamel surface collapses, there is cavity formation, which can progress to involve the underlying dentin.

#### 3.2.5. The Mature Biofilm

The inter- and intra-species relationships that play out on the tooth surface define the very biofilm communities that generate organic acids, thereby exposing the dentition to low-pH environments that are conducive to demineralization. This is supported by literature reports of specific interspecies relationships that are crucial to a community structure that promotes caries [92,97,98]. Further, community interactions between local groups of microorganisms establish intricate networks of communication that regulate pH, oxygen, and nutrient availabilities. The resulting biofilm ultimately has a distinct architecture comprised of proteins, carbohydrates, and functionally structured communities with cariogenic bacteria that together define a cooperative microbiome that derives input from both host and microbe [69].

Biofilms become organized once they are adherent to the tooth surface. First, an extracellular polysaccharide (EPS) matrix housing small aggregates of bacterial cells called micro-colonies can be seen, followed by the establishment of a communication network comprised of fluids and nutrient channels. The EPS and a heterogeneous secretion of microbial biopolymers that can include proteins, glycoproteins, glycolipids, and extracellular DNA form a sort of hydrated gel that protects and nourishes the microbial biofilm community and renders it adherent [45]. The biopolymer constituents of the plaque biofilm engage in electrostatic interactions, which largely define the physical structure of the biofilm. Weakening the EPS structure is one approach to suppressing the caries microbiome [45] and is the basis of recent novel therapies.

Ultimately, the tooth-adherent microbial community reaches maturation when the secondary colonizers, called “bridging species”, facilitate the formation of interspecies coaggregation bridges, causing bacterial population dynamics to become re-organized and biofilm detachment when this reaches a critical mass [99]. It is this process of maturation that allows microbial communities to form on the surfaces of the teeth. However, it is the tenacious attachment of the community to the host dentition that ensures the microbial production of acidic byproducts and demineralization of the tooth enamel. Thus, the stages of caries development depend upon both host and bacterial contributions.

### 3.3. Culture and Genetic Analysis of the Cariogenic Microbiome

Approaches to the study of the composition of the oral microbiome have evolved over time and incorporate both culture- and molecule-based methods. Culture methods were used in the 1960s and 1970s to demonstrate the bacterial role in dental caries. Different selective media were used to enrich for the detection of *S. mutans* [100] and lactobacilli. Acidic agar [101] or broths [102] facilitated the isolation of the acid-tolerant microbiota from carious lesions. Methods for increased sensitivity for bacterial detection followed the use of anerobic methods for strain isolation [103]. The detection sensitivity for culture-based methods is ~5% of the microbiota if 40 colonies are processed for identification. Thus, culture-based studies using isolation agar have the most value for studies that focus on identifying the dominant species.

Most of our current knowledge of the complex human microbiome, including the oral microbiome, is derived from 16S rRNA gene sequencing [104,105]. 16S rRNA-based microbial profiling studies in caries date back to the early 2000s. Since this approach has a higher detection sensitivity for bacteria present in low proportions in the microbiota, its applications reveal a more diverse ecosystem than culture-based studies [106,107,108]. Gene-based studies confirmed that *S. mutans* frequently accounts for a small fraction of the bacterial community in carious lesions, even less than noted previously in culture-based analyses [65]. A consistent observation when using16S rRNA gene community profiling approaches has been a significant reduction in bacterial community diversity associated with caries and caries progression compared with caries-free sites [109,110]. This deleterious shift within the microbial community lesions likely reflects suppression of acid-sensitive species in areas of acid-mediated lesion progression.

Side-by-side analysis using genetic and cultural methods has suggested that for dental caries, culture techniques can detect most of the taxa observed by molecular methods. Similar microbiotas were observed using contemporary anerobic culture methods when isolates and clones were sequenced for bacterial identification from dentin lesions [111] and severe early-childhood caries (S-ECC) [112,113]. An anerobic culture of S-ECC highlighted the dominance of acidogenic gram-positive rod species in *Actinobacteria*, particularly in *Scardovia* (in the *Bifidobacterium* group) and *Actinomyces* [114]. Sample analysis using pyrosequencing on a 454 platform, an updated genetic method compared with previous 16s rRNA sequencing and cloning techniques, has continued to indicate under-representation of *Actinobacteria* by genetic methods compared with anerobic cultures [115]. Thus, data deriving from genetic and culture methods are both valuable for studies of the oral microbiome.

Nevertheless, the advent of massively parallel sequencing technology, known as next-generation sequencing (NGS), has revolutionized the biological sciences [116]. This NGS-based 16S rRNA sequencing approach has become a well-established, culture-free and cost-effective method that enables analysis of the entire microbial community within a sample. It also allows for the comparative study of bacterial phylogeny and taxonomy from complex microbiomes [117,118].

## 4. Caries Microbiome at Different Lesion Sites

There is no single microbiota of dental caries; rather, carious lesions differ in bacterial composition reflecting several factors, based in part on subject age, diet, and systemic health, as well as on the site of the lesion, the extent of the lesion, and how rapidly lesions progress. Studies involving sufficient participants to overcome variations based on individual differences allow description of the caries microbiota at different clinical states (Figure 2). We first describe the species in various caries lesions and then focus on the characteristics of selected caries-associated bacteria including *S. mutans*, *Lactobacillus*, *Bifidobacterium*, and *Scardovia* species. Considering the complexity of the oral microbiome and community activities, the role of individual species in disease has been challenged. Rather, there is consideration of acid production from the microbial complex as a whole, with a reduced focus on selected putative cariogenic pathogens.

### 4.1. Initial Caries

Initial caries can present as white-spot enamel lesions (WSL), which, according to the ecological plaque hypothesis, result from biofilm colonization with more acidogenic and acid-tolerant bacteria than is otherwise characteristic of health [72]. Initial caries had been studied in culture by Kilian Clarke with the first description of *S. mutans* [60]. A longitudinal study of the development of WSL in schoolchildren using anerobic culture methods 30 years later revealed detection of *S. mutans* and lactobacilli in lesions in some but not all of the children [119]. They observed the ubiquitous presence of *Actinomyces*, *Streptococcus*, *Veillonella*, and *Bacteroides* species, and, at lower proportions, *Neisseria*, *Lactobacillus*, *Haemophilus*, *Fusobacterium*, *Rothia*, *Bacterionema* (*Corynebacterium*), *Leptotrichia*, and *Eubacterium* species.

In college students, associations between *S. mutans* and caries compared with non-lesion sites required pooling together plaque from several WSL [120]. Proportions of *S. mutans* ranged from 0.0001–10% of the microbiota assayed by culture of lesions. Based on the ability of plaque to produce a low pH in in vitro assays, van Houte et al. concluded that several bacteria other than *S. mutans* and lactobacilli contributed to the cariogenic microbiome of WSL. Using an approach based on low-pH agars, similar numbers of acid-tolerant bacteria were present in health and initial caries, including lactobacilli streptococci (principally *S. mutans* and *S. oralis*), with *Veillonellae* being associated with lesions in adolescents [101].

The microbiota of WSL has also been examined in an orthodontic fixed-banding model. In adolescents with poor oral hygiene, plaque collects around the metal bands and can be associated with rapid enamel demineralization, which appears clinically as whitened zones. Consistent with the above-cited studies, *S. mutans* was detected in only a few of the developing carious lesions [121]. Using a combination of a 16S rRNA gene-based methods, the association of *S. mutans* with WSL was confirmed [122]. The lesion-associated microbiota revealed from a microarray approach included the acidogenic streptococci (*S. mutans*, non-mutans streptococci) as well as *Atopobium parvulum*, *Dialister invisus*, *S. wiggsiae*, and *Prevotella* species [122,123].

A cross-sectional model of caries using 16S rRNA sequencing and cloning and probe assays was used to compare the microbiotas of caries-free children with those of children with caries-free sites, white-spot lesions, cavities, and advanced deep dentin caries in the same mouth [106,124]. According to this model, species associated with initial lesions, in addition to *S. mutans*, included *S. sanguinis*, *S. salivarius*, *Actinomyces gerensceriae, Veillonella* species, and a *Corynebacterium* species, although differences were not all statistically significant. These were not, however, the only species detected, as almost 200 distinct species/taxa were revealed in the samples, reflecting the complex microbiota in oral biofilms [124]. Using a similar disease model, but with longitudinal monitoring to detect new lesion development, investigators reported several microbiotas associated with initial lesion development [110]. These included a major group characterized by *S. mutans* and other smaller groups characterized by either *Streptococcus sobrinus* or *Streptococcus vestibularis/salivarius* (the assay could not differentiate these streptococci). Other progressing lesions were not included in these groups. In this study, *Veillonella* species were indicators of future caries, reflecting the presence of an acid-producing microbiota as described above [110].

These findings indicate that the microbiota of initial caries is consistent with the ecological plaque hypothesis with regards to the sequence of colonization and inclusion of highly acidogenic and acid-tolerant species, which include *S. mutans* and *S. wiggsiae* (when assayed) and several other streptococci in the caries microbiome. Initial lesions are clinically reversible, and many will not progress to cavities. Many of the white-spot-lesion-associated microbiota, particularly the actinomyces, are moderately acidogenic and acid-tolerant, consistent with the microbiome at the borderline of the dynamic stability and acidogenic stages (Figure 2) of the expanded ecological plaque hypothesis. Following the recent observation that *S. mutans* can be concentrated as “rotunds” over white-spot lesions [125], it seems likely that *S. mutans* alone could be the prime species that drives enamel demineralization. To evaluate how frequently this highly localized, species-specific demineralization occurs in initial and more advanced caries, highly localized site sampling of progressing lesions will be required. If found, it would suggest that *S. mutans* is a keystone pathogen of caries.

### 4.2. Dentin Caries

Dentin caries occurs either from an extension of deep cavities in the enamel or directly when tooth roots become exposed and lack the protective benefits of an overlying enamel layer. Tooth root exposure can follow gingival recession either associated with aging or after periodontal surgery. Reduced salivary flow as a side effect of medications, following head and neck radiation treatment, or diseases like Sjögren’s syndrome can significantly impact the microenvironment of the root surface, favoring biofilm formation and other high-risk factors for root caries.

#### 4.2.1. Root Caries

Studies in the 1970s reported a complex microbiota from anerobic culture of root caries in adults [126]. Loesche, Syed, and co-workers detected *S. mutans*, *S. sanguinis* and other streptococci, *Actinomyces viscosus*, *Lactobacillus*, and *Veillonella* species in caries. Using culture isolation on acidic media, the bacterial taxa detected in association with root caries included *S. mutans*, *Actinomyces israelii*, *Propionibacterium*, *Bifidobacterium*, *Lactobacillus*, yeast, and non-mutans *Streptococcus* species, expanding the acidogenic bacteria in root caries [127,128,129,130,131]. On a newly formulated *Bifidobacterium* selective medium, the microbiome of active lesions was comprised of 8% *Bifidobacteria [Bifidobacterium dentium*, *Parascardovia denticolens*, *Scardovia inopinata*, *S. wiggsiae* (*Scardovia* genomosp. C1), *Bifidobacterium breve*, and *Bifidobacterium subtile]*, 4% *S. mutans*, and 31% lactobacilli, as a proportion of total anerobic counts [130]. In other studies, yeast species were detected in root caries in over 60% of older Chinese individuals, although yeast levels were low and included *Candida dubliniensis*, which had previously been reported in HIV subjects [132]. The dominant species cultured in root caries in several studies thus were principally acidogenic and acid-tolerant taxa.

Molecular-based methods including clonal methods combined with 16S rRNA probe analyses were used to reveal the diverse microbiota in root caries, including *S. mutans*, *Enterococcus faecalis*, *Actinomyces*, *Lactobacillus*, *Atopobium*, *Olsenella*, *Pseudoramibacter*, *Propionibacterium*, *Selenomonas*, and *Prevotella* species [133]. The authors noted that the microbial composition of lesions varied between subjects, and that no defined microbiota was consistently observed. A pyrosequencing study indicated higher levels of several acidogenic species, including *S. mutans*, several *Lactobacillus* species, *S. wiggsiae*, *B. dentium*, and *P. denticolens* as from the culture analyses, with the addition of several proteolytic *Prevotella* species [134].

Although several *Actinomyces* species have been considered pathogens of root caries from the above culture-based studies, a recent study described similar levels of *Actinomyces* gene expression in both sound and carious root biofilms. This suggests either a role for these bacteria as commensals on the root surface or their essential function in microbial metabolic pathways and the formation of an environment on root surfaces that is suitable for cariogenesis [135]. The study authors concluded that some *Actinomyces* could be cariogenic, considering their ability to survive in acidic environments and to ferment carbohydrates.

#### 4.2.2. Deep Dentin Caries

On tooth crowns, caries involving dentin occurs after breaching the overlying enamel, and the microbiota has been considered to differ by lesion depth. The microbiota from sequential excavator samples that went deeper and deeper in lesions, however, did not differ in a study with microbial analysis by 16S rRNA cloning and by anerobic culture [111]. Some differences in the microbiota were observed based on microbiology technique, although *S. mutans*, *Rothia dentocariosa*, and *Propionibacterium acidifaciens* were detected with both approaches. The major species detected in culture included the gram-positive rods *P. acidifaciens*, *Olsenella profuse*, and *Lactobacillus rhamnosus*, whereas the dominant species in the molecular cloning analysis were *S. mutans*, *Lactobacillus gasseri/johnsonii*, *L. rhamnosu*s, *Veillonella* species (mainly *Veillonella dispar*), and several *Prevotella* and *Fusobacterium* species. Similarly, no differences in microbiota were observed by lesion depth in a 16S rRNA probe hybridization study of adolescents [136]. In this latter study, dentin species included *Fusobacterium nucleatum*, *Atopobium genomospecies* C1, *Lactobacillus casei*, veillonellae, *S. mutans*, bifidobacteria, and *Rothia dentocariosa. S. mutans* was identified in 44% of lesions, whereas other streptococci were observed more frequently [136]. Lack of difference by lesion depth was observed in a more recent study using a shotgun metagenomic sequencing approach [137]. Overall, many of the species were similar to those detected in root caries, with the addition of more proteolytic species, including *F. nucleatum*.

The complexity of dentin caries was clarified, in part, in a report describing different microbial combinations: one complex dominated by lactobacilli, another complex dominated by prevotellae, and a third complex by a combination of *Lactobacillus* and *Prevotella* species [138]. Other species detected included *Selenomonas*, *Dialister*, *F. nucleatum*, *Eubacterium*, *Lachnospiraceae*, *Olsenella*, *Bifidobacterium/Scardovia*, *Propionibacterium*, and *Pseudoramibacter.* This report suggested that the environment deep in dentin harbors a diverse microbiota with different dominating taxa, which may or may not be outcompeted by *Lactobacillus* species.

The microbiome of dentin caries was subsequently described according to lesion pH, with a lower pH correlating with increased lesion activity [139]. This is consistent with acid as a major driving pathogenic mediator in dental caries. In the latter Kuribayashi report, higher levels of lactobacilli were detected in the more acidic lesions and, in a later study, lower microbial diversity was associated with increased dentin acidity [55]. In contrast, at less acidic levels, the microbiota was likely to be more diverse and dominated by *Prevotella* species [140]. These observations indicate that the microbiota of aggressive caries is less diverse than chronic lesions, reflecting suppression of the acid-sensitive bacteria at very low pH levels.

Microorganisms involved deep in dentin are less likely to be dependent on fermentable carbohydrates (acidic stress) than in coronal caries, but more sensitive to microenvironmental stress when at the exposed sites of root surfaces [73]. The presence or absence of specific nutrients, oxygen, host defenses including secretory immunoglobulin A, lysozymes, lactoferrin and defensins, antimicrobial glycoproteins, peptides, and antibiotics from medication therapy in saliva and gingival crevicular fluid (GCF) can play significant roles in determining the composition of microbial colonization and composition on the root surfaces and susceptibilities for caries [77,141].

Furthermore, it is likely that dentin-caries-associated bacteria may be involved in caries progression via different mechanisms, either by demineralization by acidogenic species or proteolytic action deeper in lesions by *Prevotella* and *Fusobacterium* species [73] (Figure 2). Protein-degrading bacteria isolated from root caries included *Prevotella*, *Actinobaculum*, and *Propionebacterium* species [131], which were associated with final pH values of ~5.0, higher than that of acidogenic species. The proteolytic activity of these taxa likely explains the higher pH observed in deep carious lesions [140]. The heterogeneity of the microbiome of carious dentin compared with crown/coronal tooth sites was observed from metatranscriptome experiments where a more diverse microbiota was observed when gene expression profiles were mapped back to species [142] (Figure 3). Moreover, species in the dentin had a wide range of metabolic capabilities, reflecting differences in the ecology of dentin caries compared with that on the tooth surface.

### 4.3. Caries Microbiome of Aggressive Lesions

Examining the microbiota of aggressive caries presents an opportunity to identify the species most likely involved in disease progression. While active caries can occur in adults, particularly devastating is the rapid destruction of the dentition in young children. Studies in the 1970s from Walter Loesche’s laboratory using anerobic culture revealed a complex microbiota in children with lesions extending into the dentin [100]. *S. mutans*, *S. sanguinis*, other streptococci, *Actinomyces viscosus*, *Lactobacillus*, and *Veillonella* species were identified from caries-associated crown or dentin caries. *S. mutans* was implicated as a major player in caries [65]. Another population of children with severe “nursing” caries were found to harbor a higher diversity of *Lactobacillus* species, with increased numbers of *S. mutans* and lactobacilli detected in association with more sugary diets [143]. Studies since the 1970s have documented specificity in the acquisition of *S. mutans* in a child from the mother where both have dental caries [144,145,146].

In the U.S., nursing bottle caries is recognized under “severe early childhood caries” (S-ECC). While generally attributed to *S. mutans* infection, studies since the 2000s have assigned greater diversity and differing microbiotas to the active lesions. Using molecular methods, a seminal study performed in the Griffin laboratory reported the detection of significantly higher levels of *S. mutans* in S-ECC than in caries-free children [106]. Other major caries-associated species included *Actinomyces*, particularly *Actinomyces gerencseriae* in the earlier stages of the disease, an unnamed *Bifidobacterium* species (now recognized as *Scardovia wiggsiae)*, veillonellae, and non-mutans streptococci. Anerobic culture analysis indicated significant associations of *S. mutans* and *S. wiggsiae* with S-ECC [113]. Further, *S. wiggsiae* was detected in a proportion of caries-affected children without *S. mutans*, suggesting this *Bifidobacterium*-like species could be an alternate caries pathogen to *S. mutans.* Computer modeling of their microbiota, diet, and clinical characteristics distinguished S-ECC children with a high frequency of *S. mutans*, *S. sobrinus*, and *S. wiggsiae* and caries progression from other children, suggesting that these species were associated with more aggressive caries [147].

Anerobic culture isolation on acidic agar enhanced detection of *S. mutans* and *Scardovia wiggsiae* in S-ECC, but higher proportions of acid-tolerant taxa, including *S. thermophilus*, *S. intermedius*, *V. atypica, V. parvula*, and *V. dispar*, were detected in caries-free children, suggesting that acid tolerance by itself, even in acidogenic species, is not an indicator of a caries pathogen [113]. Further, there were lower microbial counts on an acidic agar medium compared with a pH-neutral agar [148], suggesting that, in part, the acidic environment of carious lesions accounts for the lower microbial diversity in disease, as reported from 16S rRNA gene community profiling approaches in S-ECC compared with caries-free children [109].

Severe early childhood caries can disproportionately affect American Indian (USA) and First Nation (Canada) children [149,150]. The caries-associated microbiota includes a very high prevalence and high levels of *S. mutans* [151], with differences in the genetic diversity of *S. mutans* compared with other children in the local area [152]. To further investigate the microbiome of aggressive caries, advanced carious lesions observed in Romanian adolescents who had very limited dental care were compared with a Swedish population with lower caries experience [153]. The high-caries population demonstrated a stronger association with *S. mutans* and *S. sobrinus* and caries, whereas the low-caries adolescents harbored a more diverse microbiota that included non-mutans *Streptococcus* and *Actinomyces* species. This provides more evidence for an association between mutans streptococci and aggressive caries. A subsequent report focused on the lower caries incidence in Swedish children and *S. mutans* detection in relation to caries. As in the previous report, adolescents positive for *S. mutans* had higher caries scores than those without *S. mutans* [154]. Microbial communities differed between the *S. mutans*-positive and -negative groups. Species in the *S. mutans*-positive groups included *Actinomyces* sp. HMT 448, *S. wiggsiae*, *Stomatobaculum longum*, and *Veillonella atypica*, most of which are highly acid-tolerant [114]. In the group with low or no *S. mutans*, additional taxa were evident, such as *Actinomyces*, *Dialister*, *Fusobacterium*, *Neisseria*, *Peptostreptococcaceae*, *Tannerella*, and *Treponema*, which were associated with low caries activity. The latter less acid-tolerant group of bacteria, including the proteolytic species involved in caries progression [131], fits the ecological plaque hypothesis extended to dentin caries [73].

A model of microbial population changes as they might relate to aggressive caries was assessed by monitoring the demineralization of dentin and enamel sections worn in the mouths of volunteers for 20 weeks [155]. More notable among the demineralization-associated species were *S. mutans* and several *Lactobacillus* species, including *Lactobacillus gasseri*, *Scardovia inopinata*, and *Rothia dentocariosa*. Since *S. inopinata* was the only *Scardovia* strain used as a reference, it is possible that the *S. inopinata* identification could have been *S. wiggsiae*. Importantly, these findings indicated that active carious lesions are linked with more acidogenic and acid-tolerant species in the microbial community, including those associated with aggressive caries.

Studies centered on the more rapidly progressing forms of dental caries thus revealed strong involvement of the more acidogenic and acid-tolerant members of the microbiome compared with less aggressive disease. These observations suggest that the enigma of caries with and without detection of *S. mutans*, may, in part, be related to the activity of the clinical disease. Acidogenic members belonging to the cariogenic microbiotas of less aggressive disease may not include *S. mutans*, but rather may include other species of *Streptococcus*, *Actinomyces*, and other gram-positive rods (including *Rothia*, *Atopobium* and *Corynebacterium*), although the cariogenic potential of many of these taxa singly or in combination have not been adequately explored.

### 4.4. Yeasts and Dental Caries

Other oral microorganisms that participate in biofilm formation and cariogenesis include species of the dimorphic fungus *Candida*, which, in its yeast form, colonizes monospecies biofilms comprised of *S. mutans* in vitro [156]. *Candida* species are detected in both health and disease [157], with *C. albicans* colonizing between 50% and 70% of caries-free individuals [158]. In the healthy oral microbiome, *Candida* species have been shown to co-aggregate with *S. gordonii* and *S. oralis*, but not with *S. mutans* [159]. However, once the local pH drops below healthy levels and sucrose is introduced into the environment, a strong adhesive interaction between the *Candida* species and *S. mutans* can be facilitated by *S. mutans*’ glucosyltransferase (Gtf) exoenzymes, which bind directly to the outermost protein layer of *Candida* [156,160]. Hence, the accumulation of acid near the dentition created by the pioneer colonizers, and the addition of sucrose to the local environment, can initiate *S. mutans* colonization and *S. mutans*–*Candida* interactions.

The presence of *C. albicans* in *S. mutans* biofilms can enhance the ability of both species to metabolize sucrose [156,161], thereby enhancing the fitness of both organisms within the biofilm community. In experimental animals, co-infection by *C. albicans* and *S. mutans* led to increased levels of colonization and rampant carious lesions [156]. Co-cultivation of *S. mutans* and *C. albicans* in biofilms revealed changes in the gene expression of *S. mutans*, with an increase in carbohydrate metabolism, which could explain the increased severity of caries observed in dual infections in vivo [161]. A second *Candida* species, *Candida dubliniensis*, previously only isolated from immunocompromised individuals such as those with HIV/AIDS, was detected in S-ECC [162] and caries-active children [163]. Furthermore, another S-ECC study revealed *C. albicans* in association with, in addition to *S. mutans*, other acidogenic and acid-tolerant species belonging to *Lactobacillus* and *Scardovia* [164]. A meta-analysis focused on evaluating the association of *Candida* with health and caries reported that children with oral *C. albicans* were at a higher risk of caries than those without the yeast species [157].

Together, these findings point to a role for interkingdom cooperation as a risk factor for dental caries that goes beyond the observed enhancement of *S. mutans*-induced caries by *Candida*.

## 5. Principal Caries-Associated Bacteria

### 5.1. Streptococcus mutans

*S. mutans* fulfils all of the criteria for species cariogenicity proposed by Van Houte [165]. These criteria include (i) physiological cell traits including acidogenicity and acid tolerance, (ii) carious lesion production in animal models, and (iii) association with caries in humans. *S. mutans* is one of the best-characterized oral symbionts and an important player in dental caries, whose colonization of the dentition can define an early stage of caries development. The primary energy source for *S. mutans* is sucrose, which is obtained by the bacteria during host mealtimes and is broken down via the homolactic fermentation pathway [166]. However, *S. mutans* has evolved the ability to metabolize a diverse set of fermentable carbohydrates, which allows the bacterium to outcompete other secondary colonizers and reestablish the population dynamics of the developing biofilm in the aciduric stage of the ecological plaque hypothesis [72,95]. The multiple sugar metabolism system (*msm*) in *S. mutans* is mediated by a cascade of proteins, including ATP-binding cassette transporters, which allow for the conversion of oligosaccharides into chemical energy [96]. Other compounds, such as the disaccharide sucrose, are processed via the phosphoenolpyruvate: sugar phosphotransferase system (PTS) [96]. In short, *S. mutans* has multiple pathways for sugar metabolism, which, lacking in many other oral pathogens, allow the bacterium to outcompete its neighbors in the increasingly acidic oral environment. Once *S. mutans* colonizes the dentition about 4 h after a mealtime, a subsequent drop in pH can follow within an hour, thereby establishing an environment that is more conducive to *S. mutans* survival and persistence and less welcoming for the original acid-sensitive pioneer species [97]. This pH drop may be considered a direct consequence of the heightened rate at which *S. mutans* is able to metabolize dietary sugars that infiltrate the nutrient channels of the plaque biofilm.

In addition to generating chemical energy, *S. mutans* can also channel sucrose into the extracellular polysaccharide (EPS) biosynthetic pathway to promote its tenacious and now irreversible adherence to the tooth surfaces [96]. The copious quantities of extracellular polysaccharide typically generated by *S. mutans* can serve as an extracellular nutrient repository, as well as a diffusion barrier that traps lactic acid in close proximity to the tooth surface [95]. This accumulation of acid is crucial to the aciduric stage of the ecological plaque hypothesis, in which the plaque is dominated by aciduric microorganisms. Thus, it is no wonder that *S. mutans*’ obligate biofilm lifestyle allows it to thrive in the human oral cavity despite the transient environmental conditions and stressors imposed by proximate bacterial residents and their metabolites. Because the *S. mutans* biofilm is a more successful safeguard than those produced by other local microorganisms, the colonization of *S. mutans* and secretion of its biofilm metabolites not only appoints the species a leading role in the mouth but also fosters the survival of additional bacteria.

Research on caries-associated bacterial virulence factors has yielded controversial results. Comparative genomics of *S. mutans* from caries-active and caries-free children revealed genome homogeneity among isolates that ranged from 79.5% to 90.9%, and no specific genetic loci were identified for either group [167]. Differences in the putative virulence genes of *S. mutans* clinical isolates were not found to be associated with caries status [168]. Other studies, however, reported that specific virulence-associated genetic loci, such as the mutacin-encoding gene *mutA*, the adhesin-encoding genes for SpaP and Cnm, and the glycosyltransferase modulating gene *SMU.833* for *S. mutans* biofilm formation could potentially contribute to *S. mutans* cariogenicity [169,170,171,172]. These variations in *S. mutans*, along with other potential cariogenic microorganisms, illustrate the complexity of studying caries-associated microbiomes and developing effective treatments. Furthermore, while *S. mutans* has a solid association with dental caries, this species is not sufficient to explain all carious lesions, since caries frequently occurs in the absence of *S. mutans*. A considerable body of evidence has demonstrated that a range of low-pH non-mutans streptococci bacterial species, in addition to other species, are involved in caries development [165,173,174]. Other caries-associated acidogenic and acid-tolerant species include those in *Bifidobacterium*, *Lactobacillus*, *Scardovia*, and *Actinomyces.*

### 5.2. Lactobacillus Species

The genus *Lactobacillus* includes over 100 species that are widely distributed in nature, including in plants, animals, insects, and food, principally dairy products, meat, and beverages. Lactobacilli are gram-positive, non-sporulating rods, cocci, or coccobacilli that are highly acidogenic and acid-tolerant. They require a fermentable carbohydrate source for growth. In humans, *Lactobacillus* species colonize several body sites, including the gastrointestinal and urinary tracts and the vagina, although the lactobacilli demonstrate the highest species diversity in the oral cavity. While over 100 *Lactobacillus* species have been identified in different ecological niches [175], many of the lactobacilli are considered commensals or beneficial species in healthy humans [176,177]. The oral cavity harbors over 20 *Lactobacillus* species or phylotypes, 50% of which have been associated with active caries in children or adults [178].

Lactobacilli can be detected in the oral cavity as early as the first two months of life [179]. The colonization of *Lactobacillus* species in the mouths of infants and young children has been correlated with the mother’s vaginal microflora, breastfeeding, pacifier use, dietary habits, and the use of antibiotics [179,180,181]. Teanpaisan et al. reported vertical transmission of *Lactobacillus* between mothers and their children in a Thai family study [182]. Only 50% of *Lactobacillus* DNA, however, was a match between mother–child pairs for individual *Lactobacillus* strains. Early oral colonization by lactobacilli has been linked to an increased risk for dental caries in young children [183], and oral *Lactobacillus* counts are correlated with a higher prevalence and increased severity of dental caries [184,185]. Conversely, *Lactobacillus* species are infrequently detected in the saliva of caries-free individuals and are only rarely isolated from dental plaque on sound tooth surfaces or from saliva or dental plaque lacking *S. mutans* [185].

The association of *Lactobacillus* species with caries has varied between investigations and microbiology methods. Using selective culture-based methods, lactobacilli were detected at low proportions in the microbiota, with no clear association between individual species and caries [102]. Several molecular techniques have been developed and introduced to determine lactobacilli colonization in the oral cavity, including chromosomal DNA fingerprinting, DNA probes, polymerase chain reaction (PCR) with *Lactobacillus* genus- and species-specific primers, matrix-assisted laser desorption/ionization-time of flight mass spectrometry (MALDI-TOF), and *Lactobacillus* 16S rDNA gene sequence analysis.

Using gene probes for species detection in less advanced caries revealed several *Lactobacillus* species. Still, their associations varied in children with and without caries, and none of the species predominated in the early carious lesion [186]. In caries-active adults, on average, 2–8 distinct genotypes of *Lactobacillus* species were identified in each individual [187,188]. More recently, multiple *Lactobacillus* species were detected in children with S-ECC and in caries-free children (Figure 4A). The abundance and distribution among those *Lactobacillus* species, however, differed significantly in the children with S-ECC versus caries-free children [189] (Figure 4B).

*Lactobacillus* species were frequently detected in advanced dentin carious lesions [111,190]. As caries progresses, significant increases in *Lactobacillus* proportions have been observed [106]. Rocas et al. [191] used 16S rRNA gene sequencing to identify the bacteria in the microbiome that occupied the deepest layers of carious dentin, 42.3% of which were *Lactobacillus* species, followed by *Olsenella* (13.7%), *Pseudoramibacter* (10.7%), and then *Streptococcus* (5.5%). Half of the advanced caries lesions were dominated by lactobacilli, comprising 63% to 96% of the bacterial sequences in these samples (Figure 5) [191]. Caufield et al. reported that the dominant species in both adult and childhood caries included *L. fermentum*, *L. rhamnosus*, *L. gasseri*, *L. casei/paracasei*, *L. salivarius*, and *L. plantarum*. The less prevalent species were *L. oris* and *L. vaginalis* [192]. Other clinical studies showed that once carious lesions were successfully restored, the levels of *Lactobacillus* colonization were substantially reduced [193,194].

Taken together, these observations suggest that a carious lesion could represent the primary ecological niche for lactobacilli colonization, furthering the notion that the lactobacilli–caries relationship could be species-specific [179,182,185,186].

Most *Lactobacillus* species in carious lesions cohabitate with other lactobacilli. Multiple *Lactobacillus* species can promote caries progression in dentinal lesions. The reduction in *Lactobacillus* colonization once carious lesions were successfully restored is due to the loss of *Lactobacillus’* ecological niche in the cavities [186,187,190,191,192,193,194]. Taken together with *L. fermentum*, *L. gasseri*, *L. casei*, *L.*
*salivarius*, *L. rhamnosus*, and *L. plantarum* as the most frequently detected taxa in advanced dentin caries, these species could potentially be classified as caries pathogens [187,195].

The widely divergent species and genotypes that colonize the oral cavity suggest that the natural sources of *Lactobacillus* include exogenous and opportunistic colonizers that reside outside of the human oral cavity, likely originating from food products or other fermented materials [192]. For example, of the lactobacilli detected in caries, *L. fermentum* and *L. casei* are among the species frequently detected in animals, plants, and fermented foods, consistent with a dietary origin. *L. gasseri*, *L. acidophilus*, *L. vaginalis*, *L. crispatus*, and *L. jensenii* are ordinary colonizers of the healthy vagina (Figure 6) and/or the gastrointestinal tract. *L. salivarius* and *L. rhamnosus* are commonly used as probiotics to suppress pathogenic intestinal bacteria and so are considered beneficial organisms.

The relative importance of individual *Lactobacillus* species to dental caries; their origins, transmission and colonization pathways; their diversity, stability and roles in caries ecology; and their interactions with other acid-sensitive or cariogenic bacterial species need further investigation and clarification. Clinically, the presence of *Lactobacillus* in the oral cavity has been used as an indicator of fermentable carbohydrates and a caries-inducing oral environment. Commercial chairside tests measuring salivary *Lactobacillus* counts based on selective isolation have been used in clinical studies for caries risk assessment and post-treatment evaluations [193]. Taken together, this information should help scientists and clinicians improve caries management.

### 5.3. Actinomycetaceae

*Actinomyces* and related species have been recognized as members of plaque biofilms for many decades. Current taxonomy recognizes several families and genera in the Actinobacteria class that, based on reports in the literature, are relevant to dental caries. These include species in the genera *Actinomyces*, *Rothia*, *Bifidobacterium*, *Parascardovia*, *Scardovia*, *Corynebacterium*, *Olsenella*, and *Atopobium*, the taxonomy of which can be accessed via the eHOMD database, version 3.1 (https://www.homd.org/taxa/tax_table).

Only a few species, however, have been examined for cariogenic traits as opposed to overall disease association. Specifically, *Actinomyces* were cultured from plaque and dentin samples associated with carious lesions in children [100] and *A. viscosus* was identified as the dominant species in root caries in adults [126]. While *A. viscosus* can lower the pH below 5 in glucose broth, it was not as acidogenic or acid-tolerant as *S. mutans* isolates at an initial acidic pH [197]. Moreover, in experimental animals, *A. israelii* was found to be associated with root caries, but not with coronal caries [198]. Taken collectively, and despite the taxonomy of *Actinomyces* being updated several times since the 1980s, the actinomyces continue to be consistently isolated from carious lesions.

The association of individual *Actinomyces* species with caries status supports species specificity in health and disease. Strains of 24 different *Actinomyces* species were reported as having a broad-based ability to lower pH when grown under neutral (pH7) and acidic (pH5.5) conditions [114]. When grown in glucose broth, some *Actinomyces* species, including *Actinomyces johnsonii*, *Actinomyces graevenitzii*, and several unnamed species, were capable of lowering pH < 4 and were nearly as acid-tolerant as *S. mutans* and *S. sobrinus*, consistent with the characteristics of known cariogenic bacteria. Other *Actinomyces* species, including *Actinomyces massiliensis*, *Actinomyces georgiae*, and *Actinomyces meyeri*, were less acidogenic, reaching final pH values close to pH 5, which is like the acidogenic potential of species known to reside in caries-free sites. Thus, these differences in acidogenic and acid-tolerant properties for *Actinomyces* species support the specificity of *Actinomyces* in both health and disease and that a lack of species differentiation will obscure caries associations.

Bifidobacteria have been associated with dental caries since the early 1900s, including in a study of childhood caries in 1917 [58]. Bifidobacteria isolated from root caries were highly acidogenic [127], although cariogenicity testing in experimental animals failed to show significant caries induction, which was attributed to poor colonization in the animals tested [199,200]. The current taxonomy of the bifidobacterial group recognizes several species that colonize the oral cavity, including species in the genera *Scardovia*, *Parascardovia*, and *Alloscardovia* [113,201]. *S. wiggsiae* was significantly associated with S-ECC in young children based on molecular analysis of the oral clone CX010 [106] and an anerobic culture study [113]. A selective medium was used to isolate several *Bifidobacterium* species from occlusal lesions in both children and adults, including *B. dentium*, *P. denticolens*, *Scardovia inopinata*, *S. wiggsiae* (*Scardovia* genosp. C1), and *Bifidobacterium longum* [201]. The results of selective isolation experiments also revealed a strong association of bifidobacteria with increasing root caries severity [130].

*Bifidobacterium* species have been associated with clinical models of disease progression. *Scardovia inopinata* was associated with the demineralization of enamel and dentin sections worn in oral appliances [155]. White-spot initial carious lesions in adolescents with fixed orthodontic appliances were associated with *S. wiggsiae* in cross-sectional studies [122] and with *S. wiggsiae* and *B. dentium* in longitudinal monitoring [123]. Further, higher proportions of bifidobacteria were isolated from caries-active than caries-free children [202]. These findings implicate *Bifidobacterium* species as playing a larger role in more advanced carious lesions than in early lesions. Furthermore, higher proportions of children with advanced caries had *Bifidobacterium* species and *S. wiggsiae* in saliva compared with initial caries and caries-free children [49]. Comparison of bacteria from a metatranscriptomic analysis of dentin caries detected higher levels of *S. wiggsiae* and *B. dentium* compared with coronal caries and caries-free children [142] (Figure 3).

Clinical and in vitro studies support a symbiotic relationship between *Scardovia* and bifidobacteria with *S. mutans*. For instance, in children’s saliva samples, there was a positive association between *S. wiggsiae* and *S. mutans* [49]. Analysis of adolescents with lower and higher levels of *S. mutans* revealed an association of *S. wiggsiae* with *S. mutans* in association with more aggressive caries [154]. *Bifidobacterium* species did not form biofilms as a single species when inoculated onto glass slides [203]. This lack of biofilm growth may in part explain why it was difficult to implant bifidobacterial [199] or *S. wiggsiae* [200] in animal models for cariogenicity testing. In vitro biofilms did, however, form when strains were co-infected with *S. mutans* [203], and, for *P. denticolens* and *Scardovia inopinata*, lower pH values were achieved in dual-species culture than when either species was cultured alone. In other in vitro assays, *S. wiggsiae* strains were found to be as acidogenic and acid-tolerant as *S. mutans* [114].

These findings reflect a resurgence in interest in *Actinobacteria*, especially in *Scardovia*-related species. Findings from dual-species analysis indicate the importance of considering the action of the whole microbial community rather than relying on the activity of individual species to fully understand their cariogenic potential.

## 6. Microbiome: Beyond Microbial Composition and the Oral Microbiome: Functional Genomics

### 6.1. Metagenome

Although 16S rRNA gene-based microbial profiling can reveal the details of microbial composition associated with health or disease, this information does not readily translate into the genetic potential and functional capacity of microbes [204,205]. This is particularly important for the study of dental caries. It has been shown that the composition of the microbial consortia associated with health and caries status varies significantly between individuals and between sites, and that different bacterial consortia may be responsible for caries pathogenesis [72]. Thus, there are limitations in taxonomy-based studies that can allow different bacterial consortia to perform similar functions due to the large redundancy and plasticity of microbes [206]. Meanwhile, accumulating evidence indicates that genomic makeup and the genetic and metabolic potential of different strains within the same species (based on 16S rRNA genes) can display high variability at the strain level [169]. Thus, species-level identifications may not provide adequate information for identifying the pathogen or understanding the disease process.

Metagenomics is a DNA sequencing approach that allows for the study of the genetic material recovered directly from environmental or host-associated samples. In metagenomics, whole-genome shotgun (WGS) sequencing is used. That is, entire DNA samples are randomly sheared by a “shotgun” method and the resulting short fragments are sequenced by NGS [207]. The comprehensive sequences can then be analyzed to obtain either bacterial profiles based on 16S rRNA genes or genomic profiles based on whole genomes [208,209]. Metagenomics not only provides more precise taxonomic diversity and phylogenetic composition data associated with specific health/disease status due to the longer 16S RNA gene sequences obtained, it also offers insight into the genetic potential and patterns of the metabolic modules/pathways of complex microbial communities, which have been especially informative in studies of periodontal and caries etiology [99,210,211].

Using metagenomic sequencing analysis, Xie et al. report that the predominant functional categories within a healthy human plaque microbiome include the metabolism of carbohydrate (11.88% of the assigned reads), amino acids and their derivatives (7.89%), and proteins (9.34%) [212]. Furthermore, about 2.8% of the total predicted protein-encoding genes were involved in antibiotic resistance and toxic compound tolerance. Among these were resistance genes for the major classes of antibiotics, such as β-lactams, aminoglycosides, fluoroquinolones, and the peptide antibiotic bacitracin, as well as genes for general multidrug or heavy-metal resistance functions, such as efflux pumps.

Comparison of metagenomic DNA sequence data allows the identification of gene functions or metabolic pathways that are over-represented in, and thus positively associated with, microbiomes related to health or caries. For example, a direct comparison of metagenomics DNA sequence data derived from caries-free and caries-active subjects will likely reveal those genes and functional pathways that promote species adaptation and strain-level niche exploitation in an acidic microenvironment. The genes or metabolic pathways crucial for these adaptations can be revealed as “enriched” in caries-active compared with caries-free microbiomes.

Using comparative metagenomics, Belda-Ferre and colleagues identified more genes for mixed-acid fermentation and DNA uptake in microbiomes associated with two subjects with active caries compared with two healthy individuals [213]. Analysis of the metabolic potential of the caries metagenome showed that samples from diseased individuals tended to cluster together, suggesting similar sets of functions were present in their metagenomes. Interestingly, the oral microbiomes from healthy subjects included significantly enriched genes involved in the biosynthesis of antibacterial peptides, such as bacteriocins; periplasmic stress-response genes like *degS* and *degQ*; capsular and extracellular polysaccharides; and bacitracin stress-response genes; while samples from caries-active subjects had a high frequency of genes involved in functions such as iron scavenging and oxidative and osmotic stress response.

The results of these metagenomic studies suggest that specific metabolic genes and pathways are associated with oral diseases, although more in-depth and functionally informative analyses are required to confirm these conclusions. Some of these genes may have the potential to become diagnostic markers.

### 6.2. Metatranscriptome

Although a metagenomic approach can reveal the total genetic potential of a microbial community, it cannot elucidate which genes are being actively expressed in real time. The human oral microbiome exists in a continuously changing environment where pH, organic carbon, and oxygen levels fluctuate on a hundred-fold or even a thousand-fold scale within minutes [214,215]. All of these dynamic environmental changes can have varying impacts on the metabolic activity of different bacterial species in the community and so affect the genes that are actively expressed by that community. Thus, detecting and analyzing the gene transcripts of a microbial community offers real-time information that can help detect the metabolically active microbial members and identify the genes expressed under a given set of conditions.

Metatranscriptomic analysis characterizes the gene expression profiles of entire microbial communities based on the sets of transcripts that are synthesized under diverse environmental conditions. Instead of addressing the question “what are oral microbes capable of doing?”, as in metagenomics, metatranscriptomics offers answers to “what are oral microbes actually doing?” [216,217].

Using a metatranscriptomic approach, Simon-Soro et al. identified the RNA-based, metabolically active bacterial compositions of carious lesions at different stages of disease progression in an effort to provide a list of potential etiologic agents for tooth decay [218]. They demonstrated that the microbiota associated with dental caries in adults is highly complex, with each sample containing between 70 and 400 metabolically active species. The compositions of these bacterial consortia varied among individuals and between caries lesions in the same individuals. Enamel and dentin lesions also had a different makeup of metabolically active microbes. In sum, their data confirmed that the etiology of dental caries is likely tissue-dependent and that the disease has a clear polymicrobial origin. The relatively low proportion of *S. mutans* detected in the caries sites examined indicated that this bacterial species can be present as a minority, leading the authors to question the importance of *S. mutans* as the main etiological agent in dental caries. Alternatively, these observations implicate *S. mutans* as a keystone pathogen that, despite its low abundance at the caries site, could remodel the caries microbiota to favor dysbiosis [219] and form concentrations of *S. mutans* above localized sites of demineralized enamel [125].

A study of the metatranscriptome of progressing early childhood caries lesions, however, indicated a clear positive association of genes mapping to *S. mutans* from coronal (enamel) and dentin lesions compared with caries-free children [142] (Figure 3). While there was considerable variability in microbial gene expression within the clinical groups, similar to that observed in adults [218], total enzyme expression was higher in dentin caries compared with coronal caries and caries-free samples [142]. Further, there was greater diversity in the operational taxonomic units (OTUs) that genes from dentin mapped to than in those from coronal caries, although sample sizes were smaller from dentin than from coronal sites. Functional profiling revealed higher levels of alcohol dehydrogenase from *Neisseria sicca* and choline kinase from several non-mutans streptococci in caries-free samples; hence, both taxa are associated with healthy plaque. The expressed sequences in coronal (enamel) caries from *S. mutans* were mainly derived from DNA ligase genes, suggesting increased metabolic activity in that species. A wider range of gene expression activity was observed in dentin than in coronal caries, including expression of uracil DNA glycosylase from *S. wiggsiae* and urease from *A. naeslundii.* Moreover, the higher levels of gene expression patterns mapping to *S. wiggsiae* in dentin caries compared with coronal caries, suggest *S. wiggsiae* could be a major player in caries progression in dentin. Overall, metatranscriptomic analysis revealed some differences in the enzyme/metabolic activities that were expressed in microbiomes associated with dentin caries, coronal caries, and caries-free samples, respectively [142].

Metatranscriptomic studies thus revealed marked differences in bacterial composition and metabolism between coronal and dentin caries, suggesting that the microbial communities in enamel lesions exhibited different functions than those of more advanced, dentin cavities, where bacteria appear to be specialized in utilizing sugars associated with dentin tissue and degrading proteins. These data support the hypothesis that dental caries is not a single disease but a tissue-dependent process with different etiologies, as discussed above [99] and consistent with the ecological hypothesis for dentin caries proposed by Takahashi and Nyvad [73].

Using RNA-seq to perform global gene expression analysis of the dental plaque microbiota derived from 19 twin pairs that were either concordant (caries-active or caries-free) or discordant for dental caries [220] revealed similarities in gene expression patterns. Thus, this twin-pair study allowed an assessment of the relative contributions of human genetics, environmental factors, and caries phenotype on the microbiota’s transcriptome. The results revealed transcripts encoding functions related to monosaccharide and disaccharide metabolism, accounting for a significant portion of the dental biofilm transcriptome (around 15% of the total) [220]. Transcripts encoding functions associated with disaccharide metabolism were more prevalent than those associated with monosaccharide metabolism by a factor of two. Interestingly, transcripts encoding functions related to antibiotic resistance and tolerance of toxic compounds were also expressed in the oral biofilm, while genes involved in bacteriocin expression and acid stress represented only a small fraction of all transcripts across all individuals. Furthermore, correlation analysis of transcription identified several functional networks, suggesting that inter-personal environmental variables may co-select for groups of genera and species [220].

By applying a novel computational framework (metaModules) for the automated identification of key functional differences between health and disease-associated communities, May et al. [221] were able to identify key functional subnetworks that are relevant to dental caries, as noted in the metatranscriptomic data obtained by Peterson et al. [220]. Their results showed that the health-associated KEGG Orthology groups (Kos) were mainly constituents of the pathways associated with amino acid biosynthesis and pyrimidine, purine, and pyruvate metabolism, as well as glycerophospholipid metabolism. The caries-associated subnetwork included nine Kos from the pathways of the phosphotransferase system and fructose/mannose metabolism, supporting the notion that microbial carbohydrate acquisition and catabolism are important in dental caries etiology [96].

In sum, metatranscriptomic analysis allows one to gain insight into the genes that are actively expressed in complex bacterial communities, enabling the elucidation of dynamic functional changes that govern the microbiome’s functions in given contexts, its interactions with the host, and the functional alterations that accompany the transition of a healthy microbiome into a dysbiotic one.

### 6.3. Multi-Omics

With the advancement of sequencing technologies, protein and small-molecule analyses, and the development of new bioinformatics tools, studies of host-associated microbiomes, including the caries microbiome, have entered a new era. Investigators are now capable of generating information on the functional activity of whole microbial communities. Metagenomics, which entails sequencing the entirety of the DNA from a given sample, can provide information not only about which organisms are present, but also their functional potential, through analysis of metabolic pathway genes and the use of protein-coding sequence databases. Meanwhile, current investigations in the oral cavity and beyond that combine metatranscriptomics and metagenomics have shown that, in many cases, the gene sets that are expressed are more important in predicting health compared with disease than the species that are present [222,223]. Furthermore, meta-proteomics, metabolomics, and small-molecule analysis have also been applied to study in vivo [224,225] and in vitro [226,227] multispecies oral microbial communities to complement sequencing data by providing higher-level functional information. Combining systems-level analyses that generate functional information with knowledge of abundance and spatial structure can provide invaluable insights into the intricate microbial network of the oral microbiome, its relationship to the human host, and its role in caries pathogenesis.

## 7. Treatment of Caries as a Microbial Disease

Treatment approaches for dental caries have evolved, with an increased understanding of the cariogenic microbiome and the development of new therapies. Since dental caries, however, has a multifactorial etiology with dietary considerations in addition to bacteria and host factors, successful and lasting therapy requires attention to all these factors. The most effective antibacterial therapy is doomed to relapse in the absence of diet management. Treatment of dental caries has incorporated suppressing the cariogenic bacteria and increasing the tooth’s resistance to demineralization. The most frequent approach for caries prevention, good plaque control through oral hygiene, while effective [228], can be difficult for individuals to achieve. Thus, additional preventative measures are usually needed, especially in high-risk populations or once carious lesions have developed.

Earlier approaches to caries treatment related to the microbiota focused on reducing the colonization and transmission of caries-associated bacteria using antimicrobials and fluoride-containing formulations. More recent treatment measures have focused on rebalancing the dysbiotic caries microbiome with specific therapies targeting either *S. mutans* alone or the acidogenic, acid-tolerant microbiome with the goal of re-establishing a symbiotic microbiome compatible with health. Ecological approaches have also been proposed to avoid the development of a dysbiotic community associated with caries [229]. Highlights of these approaches are described below.

In studies in the 1970s, topical applications of broad-spectrum antimicrobials, such as kanamycin and vancomycin, were used with the goal of eliminating *S. mutans* and disinfecting carious lesions before restorations were placed [230,231]. Suppression of *S. mutans* levels in tooth pits and fissures by antimicrobial agents, however, was short-lived because agents failed to penetrate the plaque biofilm or tooth demineralized zones. Thus, this approach was ineffective in reducing counts of target species in carious lesions over time. In some cases, there were significant increases in the proportions of *S. mutans* in the plaque community due to the non-specific bacterial reduction of other species, and caries progressed twelve months after treatment [230]. Other therapeutic agents evaluated included chlorhexidine (CHX), povidone-iodine (PVP-I), casein phosphopeptide-amorphous calcium phosphate (CPP-ACP), xylitol [232], arginine deiminase system (ADS) [233,234], tea- and cranberry-derived polyphenols [235], and blue light [236]. In clinical studies, however, these antimicrobial approaches have had ambiguous results in suppressing cariogenic bacteria or changing oral microbial ecology [232,237].

### 7.1. Fluoride and Anticaries Activity

#### 7.1.1. Fluoride as a Preventative Anticaries Agent

Fluoride is the most frequently used agent for caries prevention [238]. The widespread use of fluoride in water, toothpaste, gels, varnishes, mouthrinses, tablets, drops, milk, and salt for caries prevention in topical or systemic formulations has been an effective and significant contributing factor to the remarkable decline in tooth decay in the decades since it was introduced. Community-based fluoridation in drinking water (≥0.7 parts per million F) represents a significant public health achievement of the 20th century for its efficacy in preventing dental caries [12] that has been consistently supported by scientific evidence worldwide. A study report of 2018 estimated that the preventive fraction for US children and adolescents was 30% (95% CI 11–48%) for primary teeth and 12% (95% CI 1%, 23%) for permanent teeth [239]. This cost-effective community-based approach is dependent on access to drinking water systems and public health policies at national, state, or local governmental levels [240]. Use of toothpastes or mouthrinses containing sodium fluoride (NaF), amine fluoride (AmF), acidulated phosphate fluoride (APF), and stannous fluoride (SnF_2_), among others, reduces plaque accumulation on tooth surfaces [241] and lowers the cariogenicity of dental plaque [242]. Two mechanisms described as underlying fluoride anticaries activity—antimicrobial action and promoting enamel hardness via chemical reaction—will be described in the following two subsections.

#### 7.1.2. Antimicrobial Properties of Fluoride

Fluoride’s anticaries activity as an antimicrobial has been known for several decades [243]. Elemental fluoride can inhibit bacterial carbohydrate metabolism and extracellular polysaccharide (extracellular polymeric substances, EPS) production and thus reduce bacterial adherence and the growth of cariogenic bacteria [244]. The targets of action involve (1) sugar transport via the phosphoenolpyruvate-protein (PEP) phosphotransferase system; (2) sugar uptake via the protonmotive force (PMF)-associated system; (3) proton-extruding ATPase (H^+^-ATPase); (4) biosynthesis of macromolecules related to bacterial cell division; and (5) lactate/proton efflux [245]. Fluoride also modifies the plaque ecosystem and influences bacterial interactions and composition within the plaque community [246].

In a mixed oral species culture model that included *S. mutans*, *S. sanguinis*, *S. oralis*, *Actinomyces naeslundii*, *Neisseria subflava*, *Veillonella dispar*, *Fusobacterium nucleatum*, *Prevotella nigrescens*, and *Porphyromonas gingivalis*, 10 days after exposing to glucose with NaF, proportions of *S. mutans* were significantly reduced to <3% of the species mix, compared with the culture without fluoride (Figure 7). This study suggests that fluoride exerts antimicrobial activity against *S. mutans* by inhibiting critical metabolic processes (direct effect) and reducing acid production (indirect effect) in biofilms [247]. A later report showed that NaF inhibited *S. mutans* more than lactobacilli depending on the Lactobacillus species and NaF concentration [248].

Clinically, long-term exposure to fluoride reduced the salivary levels of mutans streptococci, lowered caries scores among school children [249,250], and prevented enamel demineralization [251] and root caries among adults [4]. In other studies, minimal changes in the microbiota were observed on fixed orthodontic appliances following fluoride mouthrinses containing 100 ppm AmF and 150 ppm SnF_2_ [252]. Further, short-term changes in dental plaque accumulation were observed in pediatric patients after a topical treatment with 0.4% SnF_2_ or 10% povidone iodine (PVP-I) plus 5% NaF varnish. Further, there was a minimal antimicrobial effect on the microbial communities of children at high risk for caries, suggesting that bacteria within dental plaque were resilient to these treaments and maintained the community structure and ecosystem [253,254].

Fluoride varnish (FV) is a frequently used formulation in clinical practice. FV applied topically can form globules with fluoride and calcium elements on enamel surfaces [255]. Formulations of FV include those with 5% NaF, 0.9% difluorosilane (SiH_2_F_2_), and 6% NaF with 6% CaF [238]. In three studies, including a review of 22 trials involving 12,455 children and adolescents worldwide, professionally applied FV (two to four times a year) was associated with a reduction of 18% to 43% in caries incidence in permanent teeth and a 37% reduction in the primary teeth of children and adolescents compared with placebo or no intervention [255,256,257,258]. Administration of FV containing 5% NaF every three to six months is the recommended treatment for children with an increased caries risk [259]. For children with early childhood caries, however, clinical studies on FV antimicrobial efficacy have yielded mixed results. Topical fluoride application alone, or in combination with other antimicrobial agents, was reported to lead to only a short-term reduction in mutans streptococci or lactobacilli levels in the saliva and dental plaque [260,261]. Other studies, however, showed no direct correlation between fluoride release and antimicrobial activity [262,263], and, therefore, more intensive regimens have been proposed [264].

Silver diamine fluoride (SDF) to treat caries is based on the activity of both silver and fluoride. For centuries, elemental silver (Ag) has been known to exhibit antimicrobial effects due to its properties as a heavy metal. Silver was the basis of “Howe’s solution”, used to treat rampant caries in children 100 years ago. The antimicrobial effect of silver compounds was established for the prevention and treatment of infections in medicine [265]. In dentistry, SDF’s caries-arrest effect was demonstrated in the late 1960s and early 1970s in Japan. Application of topical SDF, while causing black staining on treated tooth surfaces, was effective in arresting caries progression in 66% to 75% of primary teeth [266,267], 41% to 65% of first permanent molars [268,269], and 18% to 71% of root caries in elders [270].

Antimicrobial mechanisms for caries suppression by SDF include, at the cellular level, interference with bacterial amino and nucleic acid formation, alteration of cell-wall synthesis and cell division, and disabling of metabolic and reproductive functions, leading to inhibition of bacterial growth and biofilm formation [271]. An antibacterial effect of SDF against *S. mutans* growth with inhibition of dextran-induced agglutination was reported in the 1970s [272]. Subsequent studies showed that a 3.8% SDF solution could effectively suppress *Enterococcus faecalis* biofilm formation [273], which could be valuable for treating root canal infections. Knight et al. conducted a series of experiments demonstrating that application of SDF reduced *S. mutans* viability and inhibited biofilm formation, reducing the bacteria-induced demineralization of dentin and caries progression [274,275]. Chu and Mei et al. reported the antimicrobial effect of a 38% SDF solution against multi-species cariogenic complexes (*S. mutans*, *S. sobrinus*, *L. acidophilus*, *L. rhamnosus*, *and Actinomyces naeslundii*) on dentin carious lesions [276,277]. A formulation of SDF with potassium iodide (SDF/KI) reduced *S. mutans* colonization in dentinal tubules compared with chlorhexidine or other chemo-mechanical products [278]. Furthermore, SDF suppressed *S. mutans* biofilm formation on dentin caries models with a superior antibacterial effect against *S. mutans* compared with NaF and with SDF + NaF (Figure 8) [279].

#### 7.1.3. Chemical Action of Fluoride to Strengthen Teeth

Fluoride affects the enamel structure with the formation of fluorapatite, which resists demineralization and enhances the remineralization of incipient lesions compared with hydroxyapatite without fluoride. Fluoride compounds, including sodium fluoride (NaF), sodium monofluorophosphate (MFP, Na_2_PO_3_F), stannous fluoride (SnF_2_), amine fluoride (AmF), acidulated phosphate fluoride (APF), zinc fluoride (ZnF_2_), titanium tetrafluoride (TiF_4_), and ammonium fluoride (NH_4_F) have all been evaluated for cariostatic mechanisms [250,255,280,281]. Clinical studies and in vitro experiments found that NaF, SnF_2_, and MFP in dentifrice or mouthrinses had better fluoride uptake by enamel lesions, higher F¯ retention level in saliva and dental plaque, and greater efficacy than other compounds tested [282]. They, therefore, have been most frequently used in toothpaste and mouth rinses.

Several chemical reactions have been proposed for interactions between SDF and tooth structure. A combination of tooth hydroxyapatite, [Ca_10_(PO_4_)_6_(OH)_2_], silver, and fluoride ions in a 38% SDF solution resulted in the formation of an impermeable layer of silver phosphate (Ag_3_PO_4_) and calcium fluoride (CaF_2_) on the treated tooth surfaces [283]. Four decades later, deep carious lesions treated with SDF were found to have increased calcium, phosphate, and fluoride ions in caries-affected dentin [284]. Mei et al. suggested that silver particles and fluorohydroxyapatite formation in SDF-treated lesions might contribute to the increased hardness of treated tooth surfaces [285]. Silver particles were detected on the surface and in carious lesions treated with SDF, with the extent of silver penetration and intensity being correlated with the degree of demineralization of the enamel and dentin [286]. In addition to the formation of an impermeable layer of silver phosphate, calcium fluoride, and fluorohydroxyapatite in SDF-treated carious lesions, highly concentrated silver precipitation zones form around the carious lesions, thereby blocking dentin tubules, which could contribute to the physiochemical role of silver compounds in caries arrest.

Overall, topical applications of fluoride have proven to be an effective anti-caries measure. A review of data from 71 students concluded that the use of fluoride toothpastes resulted in a 23% to 36% caries reduction depending on the fluoride concentration when compared with placebo formulations [281]. The use of fluoride mouthrinses in 37 randomized controlled trials of over 15,000 children and adolescents resulted in a 27% reduction in caries increment for permanent tooth surfaces when compared with placebo or no treatment [250]. Hence, fluoride-containing toothpastes and mouth rinses are effective in school-based programs for caries prevention in children, particularly in countries with a high prevalence of caries and limited dental care resources. SDF treatment is minimally invasive, inexpensive, and time-saving in application. Therefore, it has been considered as an alternative treatment for at-risk patients when other forms of caries control are not available or not appropriate [287,288]. Nevertheless, there are limitations in anti-caries effectiveness for fluoride-containing mouthwashes, FV, and SDF, suggesting that improved treatment and prevention approaches are needed for dental caries.

### 7.2. Silver Nanoparticles

Nanoparticles containing active agents represent a new route to deliver antimicrobials in dentistry, including nano silver fluoride (NSF) and silver nanoparticles (AgNPs). These silver-containing nanoparticles are aimed at preventing cariogenic bacterial cell adhesion, attachment to tooth surfaces, and biofilm formation and maturation. These materials have greater remineralization efficacy to arrest active dentin caries and are more biocompatible than SDF without producing discoloration of treated tooth surfaces. The approximately 10 nm nano-scaled silver particles can continuously release silver ions, maintain a high contact surface with microorganisms [289], pass through bacterial cell membranes, disrupt cell processes, penetrate into dental tubules, and produce a higher antimicrobial effect against *S. mutans* and lactobacilli at lower concentrations than SDF, resulting in significantly decreased toxicity from the amount of silver ions leaching from the materials [290,291]. Acrylic resin fillings containing SF or AgNPs reduced the colonization levels of *S. mutans* and *C. albicans* [292]. Pyrosequencing of 16S rRNA gene studies showed differences in the composition of early and mature microbial biofilm communities formed on the surfaces of modified dental polymers incorporated with AgNPs [293]. Therefore, NSF and AgNPs could be alternatives to SDF as new antimicrobial anticaries agents.

### 7.3. Approaches for Caries Control Targeting the Microbiome

Development of second- and now third-generation sequencing technologies has allowed investigators to examine the full complexity of the healthy human microbiome. It is now understood to be highly diverse and contribute significantly to immune modulation, digestion, colonization resistance against would-be pathogens, and other functions [294]. This insight has led to a major paradigm shift from the concept of “one-germ, one-disease” to that of dysbiosis and polymicrobial diseases. Despite historical studies of a single pathogen per disease, the vast majority of therapeutics currently in use to treat maladies of microbial origin have a wide variety of activities, as discussed above. The adverse effects of broad-spectrum antimicrobial therapies against many diseases are now well established [295]. Furthermore, administration of broad-spectrum antibiotics may suppress not only the pathogen of interest but also wide swaths of the overall microbiota, including protective species. This leaves the treated body site prone to blooms of antibiotic-resistant pathogens or recolonization with a less-than-optimal, potentially even harmful, microbial community. More precise targeting of pathogenic species by treatment modalities that leave the remainder of the microbiome unharmed is an objective that has inspired a significant amount of research in recent years.

Historically and contemporarily, there have been several endeavors to combat dental caries by precisely targeting *S. mutans*, although none have maintained substantial traction long-term. Research investigating the feasibility of active or passive immunization against dental caries has been sporadic. Early investigations on the topic have been excellently reviewed [296]. More recent studies have explored vaccination using a recombinant P1 adhesin antigen [297], a DNA-based vaccine against glucosyl-transferases and surface proteins [298], and a glycoconjugate vaccine based on rhamnan surface polysaccharides [299]. Unfortunately, past, present, and likely future translational efforts to move anti-caries vaccine research into clinical trials face significant headwinds because caries, by itself, is not a life-threatening disease. There are currently no licensed vaccines to prevent dental caries, and, to our knowledge, only one candidate vaccine has proceeded to phase II clinical trials [300]. Conceptually, the use of bacteriophages is an attractive approach to combat *S. mutans* and dental caries, but this has received relatively little attention. Although the few phages known to infect *S. mutans* were lytic and completely eliminated viable counts from single-species biofilms, the phages demonstrated a highly stringent host specificity, which was considered a significant disadvantage, particularly considering the high intra-species diversity exhibited by *S. mutans* [reviewed in [301]]. No testing in multi-species communities or further studies has been reported.

Probiotics approaches have sought to displace wild-type *S. mutans* using *S. mutans* strains engineered for reduced pathogenesis or species which either compete with, or directly antagonize, *S. mutans* [reviewed in [302,303]]. Although more recently discovered species, such as *Streptococcus dentisani* [304] and *Streptococcus* A12 [90], hold promise compared with older candidates, no probiotic formulations have successfully demonstrated safety and efficacy in adequately rigorous clinical trials [305,306]. Two small molecules were recently reported to exhibit the ability to specifically disperse or inhibit *S. mutans* biofilms [307,308]. However, disruption of *S. mutans* biofilm alone, with minimal effect on the overall dental plaque ecology, is likely to allow rapid reestablishment of the problematic community and require constant application of the therapy. Recent work has identified two antimicrobial peptides, ZXR-2 [309] and CLP-4 [310], and a vitamin D derivative, doxercalciferol [311], with antimicrobial activities against *S. mutans*. However, the antimicrobial specificity of these molecules against *S. mutans* was not reported in these studies.

Specifically targeted antimicrobial peptides (STAMPs) were developed to address the need for precision antibiotic therapy. STAMPs are synthetic peptides consisting of a targeting domain to invoke specificity and a killing domain to exert antimicrobial action against the intended species [312]. To design a STAMP directed/targeted to *S. mutans*, a “G2” killing domain, consisting of a 16-residue segment of the well-characterized potent broad-spectrum antimicrobial peptide novispirin G10, was chosen [313]. The targeting domain selected was “C16”, which consisted of the 16 C-terminal residues of the *S. mutans* pheromone, a competence-stimulating peptide (CSP). The directed/targeted and killing moieties of C16G2 were joined by a flexible triglycine linker.

Analyses of C16G2 showed that it was capable of the targeted killing of *S. mutans* in either planktonic or biofilm settings and that it could selectively kill *S. mutans* in a three-species biofilm [313,314] as well as an in vitro, saliva-derived oral community, which consisted of over 100 species [315]. This process was accompanied by a significant increase in the relative abundance of *Streptococcus mitis* and other *Streptococcus* species associated with good dental health [315]. Further work determined that C16G2 exerted its killing effect through membrane disruption and that the cytotoxic effect of C16G2 was due to rapid killing of *S. mutans* in less than one minute of exposure, acceptably swift for application as an oral care product [316]. This STAMP was also soluble in aqueous solutions for delivery into the oral cavity in a rinse formulation. A pilot clinical trial showed that, compared with placebo, C16G2 significantly reduced the number of viable *S. mutans* in both plaque and saliva samples, decreased lactic acid production, and increased the resting pH of dental plaque [314]. Taken together, these results suggest that C16G2 is effective at selectively suppressing *S. mutans* from the dental plaque milieu and remodeling the community to one which is dominated by species that are associated with health. C16G2 is being developed into a dental product by a biotechnology company (C3J Therapeutics, Inc., Marina del Rey, CA, USA) and is currently in phase II clinical trials.

Several pH-responsive and acid-activated antimicrobial/anti-biofilm approaches have been developed to focus on acid-generating bacteria and cariogenic biofilms. Quaternary pyridinium salt (QPS) was developed by Sun’s group at the American Dental Association Foundation [317]. Preliminary analysis showed that QPS exhibits pH-controlled antimicrobial activity that selectively inhibits the growth of acid-producing bacteria within a multispecies community. Horev et al. reported pH-activated nanoparticles for controlled topical delivery of farnesol to disrupt cariogenic biofilms [318]. In this approach, nanoparticles are formed from diblock copolymers composed of 2-(dimethylamino) ethyl methacrylate (DMAEMA), butyl methacrylate (BMA), and 2-propylacrylic acid (PAA) (p(DMAEMA)-b-p(DMAEMA-co-BMA-co-PAA)) that self-assemble into cationic nanoparticles. Due to its hydrophobic core, nanoparticles could effectively carry farnesol, a hydrophobic antimicrobial. The destabilization of nanoparticle cores under acidic pH will trigger the release of farnesol, thus achieving the killing of cariogenic bacteria. Furthermore, farnesol-loaded nanoparticles effectively attenuated biofilm virulence in vivo, resulting in a reduced number and severity of carious lesions using a clinically relevant topical regimen (2×/day) in a rodent dental caries model.

Similarly, a strategy to control cariogenic biofilms using catalytic nanoparticles (CAT-NP) was developed by Koo’s team at the University of Pennsylvania [319]. CAT-NP, with peroxidase-like activity, is pH-responsive and can rapidly catalyze low concentrations of H_2_O_2_ at an acidic pH to produce free radicals to achieve simultaneous degradation of the biofilm matrix as well as the killing of matrix-enclosed bacteria. Using 1 min topical daily treatments, applicable for a clinical setting, they showed that CAT-NP in combination with H_2_O_2_ effectively suppressed caries development in a rat model. In a recent study, the same group demonstrated the potential antimicrobial specificity of ferumoxytol iron oxide nanoparticles (FerIONP) against biofilms harboring *S. mutans* through preferential binding that promotes bacterial killing via in situ free-radical generation [320]. While further testing is needed, these approaches hold the promise of being developed into oral therapeutics that, instead of targeting one specific pathogen, can achieve the targeted control of a group of bacteria displaying similar physiological properties that are relevant to disease pathogenesis, e.g., the acid production and biofilm formation of cariogenic pathogens.

While the targeted approaches discussed above have yet to bear fruit in the form of an approved therapeutic to prevent dental caries, the studies have contributed substantially to our understanding of the disease and *S. mutans* and are foundational in guiding current and future research. Dental caries remains a serious public health concern and one that could strongly benefit from a precision therapy, such as STAMPs, for targeting a specific pathogen, or pH-responsive, acid-activated antimicrobials for controlling a group of acid-producing, cariogenic, biofilm-forming bacteria, to supplement current recommended fluoride and hygienic regimens.

### 7.4. Challenges to Antimicrobial Approaches for Caries Management

The etiology of dental caries includes complex multidimensional interactions over time between acid-producing bacteria, fermentable carbohydrates, and host factors including genetics, tooth mineralization status, saliva composition, nutrition, immune response, and oral hygiene [321]. Caries risk in individuals can change to high risk at any time [322] depending on shifts in the symbiotic–dysbiotic supragingival microbiome and the natural demineralization–remineralization balance at the enamel–biofilm interface, which is influenced by sugar intake. The “cariogenic microbiota” has evolved not only to include increased numbers of acidogenic and acid-tolerant microorganisms identified in the biofilm [323] but also an extended list of functional traits and molecules that are involved in the oral microbial community and ecological system of disease [324,325].

Lastly, dental caries is a biofilm-mediated, sugar-driven, and behavior-modifiable disease. Host–microbiota–diet interactions in the disease pathophysiology bring together additional aspects of the complexity of studying the caries microbiome. Current research focuses on the human oral microbiome, metaproteome, and metatranscriptome of dental caries. A group of discriminant bacterial species and a list of bacterial and human proteins (involved in pH buffering, exopolysaccharide synthesis, protein synthesis, iron metabolism, and immune response) were used to create models to effectively separate clusters of healthy and diseased individuals [326] and to predict risk for caries onset [327]. These exploratory multi-omic studies will provide new insights into the possibility of developing cariogenic biomarkers for caries diagnosis, treatment, and prevention. Nevertheless, further large-scale clinical studies are needed to test and validate the findings and to achieve a better understanding of the mechanisms determining the interrelationship of those biomarkers.

## 8. Conclusions

We conclude that while dental caries remains an important clinical problem worldwide, our understanding of the associated microbial biofilms has greatly advanced. The microbiology of dental caries was built on the concepts of the golden age of microbiology, using the one-bacterium-for-one disease hypothesis. In caries research, studies of cariogenic pathogens have focused mainly on *S. mutans* and lactobacilli. Cultural, and especially molecular, approaches in microbiological analyses have evolved with associated conceptual changes to focus on the biofilm community rather than individual species. Health is now recognized as being associated with a balanced symbiotic community, including acid-balancing microbial strategies, which is lost in the unbalanced dysbiotic change associated with disease. The caries-associated microbiome is not characterized by a single microbiota but varies between individuals, tooth locations, and rates of disease progression, with some evidence that the most aggressive caries is associated with the highly acidogenic, acid-tolerant *S. mutans* and *Scardovia* species. Functional characterization of the caries-associated microbiome can lead to a better understanding of microbial activities at a community level. The study of the molecular function of the oral microbiome is leading to an expanding characterization of caries as it relates to community activity. Prevention and treatment approaches for caries have moved away from suppressing the whole community with antimicrobials, fluoride, and silver-containing agents, which had limitations despite many treatment successes. Modern therapeutic approaches focus on targeting the pathogenic components of the microbiome, including acidogenic species such as *S. mutans*, to achieve ecological rebalancing of the dysbiotic community. We look forward to an ever-increasing understanding of dental caries and strategies for its treatment, prevention, and final eradication.

## Figures and Tables

**Figure 1 microorganisms-12-00121-f001:**
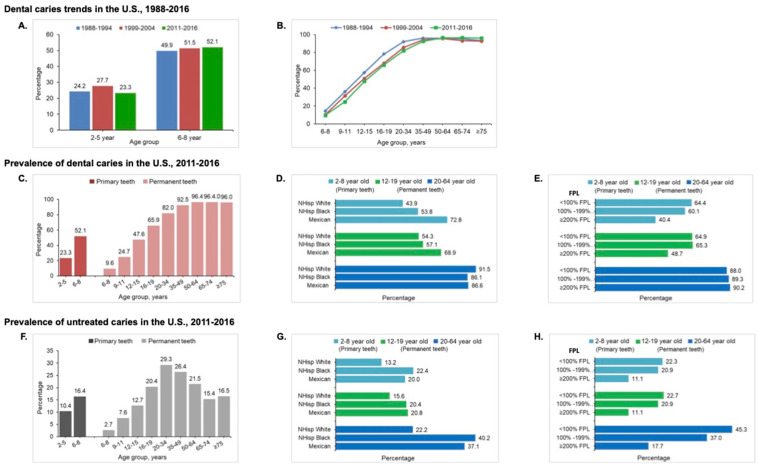
The distribution of dental caries by age, race, and ethnicity, and family income in the U.S. population. (**A**) Prevalence of caries in primary teeth. Compared with NHANES 1988–1994, caries experience slightly decreased for children 2- to 5-year-olds. However, the decrease was not observed for 6- to 8-year-old children between 1988 and 2016. (**B**) Trend in dental caries. There was a slight decrease in caries experience in the permanent dentition of young children and adolescents between 1988 and 2016. However, there are no significant changes in overall caries status in adults aged 20 to 64 years old. Caries prevalence significantly increased for the elderly aged 64 years and over. (**C**–**E**) Prevalence of caries in primary and permanent teeth in 2011–2016 by age (**C**), race and ethnicity (**D**), and family poverty status, FPL = Federal Poverty Level (**E**). (**F**–**H**) Prevalence of untreated caries by age (**F**), race and ethnicity, NHisp = Non Hispanic (**G**), and family poverty level (FPL) (**H**) in the U.S. population.

**Figure 2 microorganisms-12-00121-f002:**
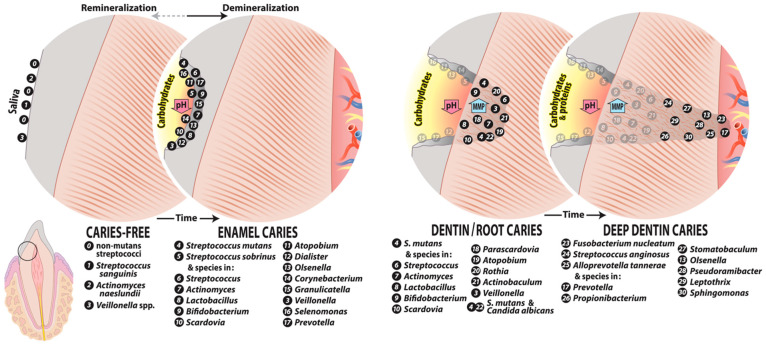
Major species identified from caries-free sites, enamel caries, and dentin/root caries. Shown are cross sections of the enamel, dentin, and root of a tooth with their associated caries-free and caries-associated microbiota. Caries-free microbiotas colonize the tooth surface as diverse communities that respond to dietary carbohydrate challenge by producing acid, favoring enamel demineralization that is neutralized by microbiome community activity that includes the production of ammonia deiminase and urease activity (remineralization). Enamel caries derives from the acid-induced enrichment of highly acidogenic species and suppression of acid-sensitive species, leading to a reduction in community diversity. Caries in dentin, including root caries at the tooth surface, generally comprises a more diverse microbiota than enamel caries with moderately rather than highly acidogenic species owing to the reduced mineral content of dentin. Caries progression deep in dentin can involve acidic demineralization in addition to increasing proteolytic activity, leading to higher pH values in deep lesions. The sequence of caries progression shown is based on Takahashi and Nyvad [73]. Species based on text references.

**Figure 3 microorganisms-12-00121-f003:**
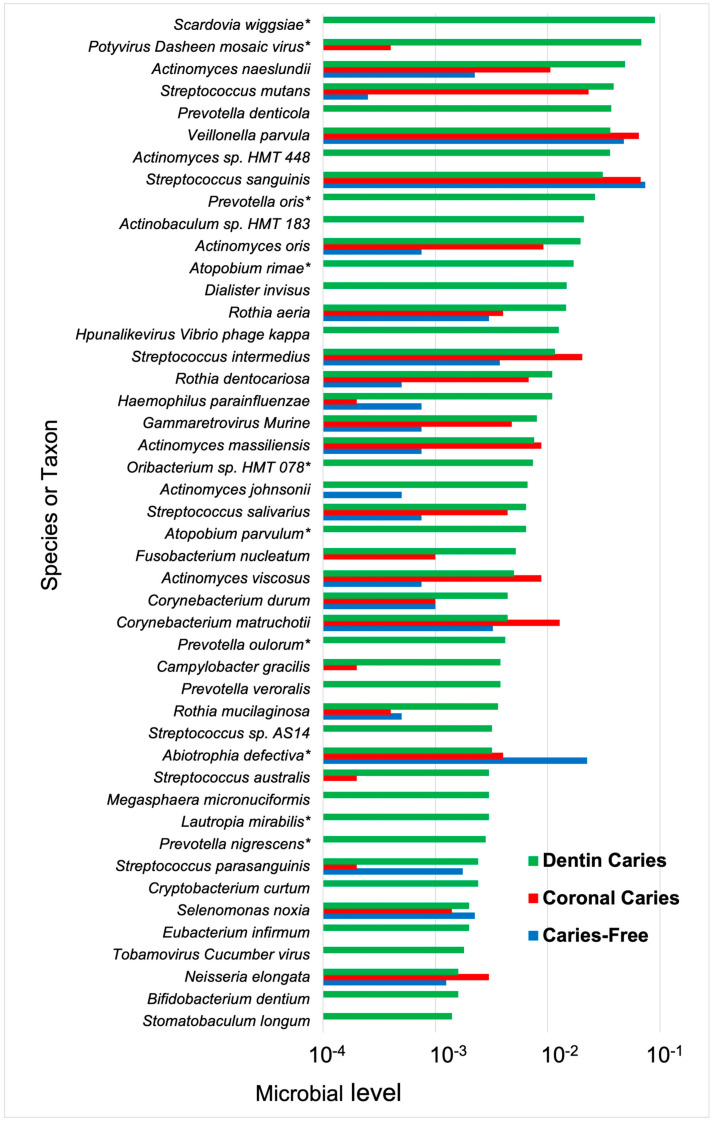
Microbial species and taxa, including viruses, detected from gene expression profiles in coronal and dentin caries. Bacterial samples were taken from coronal caries and caries-free sites and dentin caries from early childhood caries in children with progressing lesions. Functional profiling was performed on purified bacterial RNA using HUMAnN 2.0, version 0.9.9 (HMP Unified Metabolic Analysis Network). Species mapped from gene expression profiles showed the greatest diversity from dentin caries samples. Dentin caries n = 6; coronal caries n = 5; caries n = 4. * difference detection, Chi-square > 0.05. Data from Kressirer et al. [142].

**Figure 4 microorganisms-12-00121-f004:**
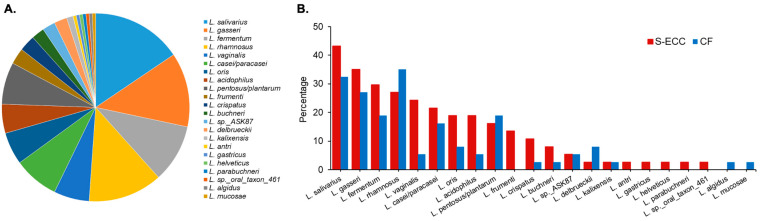
Diversity of oral *Lactobacillus* species. (**A**) Composition of *Lactobacillus* species isolated in the oral cavity of 3- to 5-year-old children (N = 74). (**B**) Distribution of twenty-one *Lactobacillus* species identified in children with severe early childhood caries (S-ECC, N = 37) and children without caries (CF, N = 37). The figure shows that the abundance and distribution of the *Lactobacillus* species were significantly different between the two groups of children [189].

**Figure 5 microorganisms-12-00121-f005:**
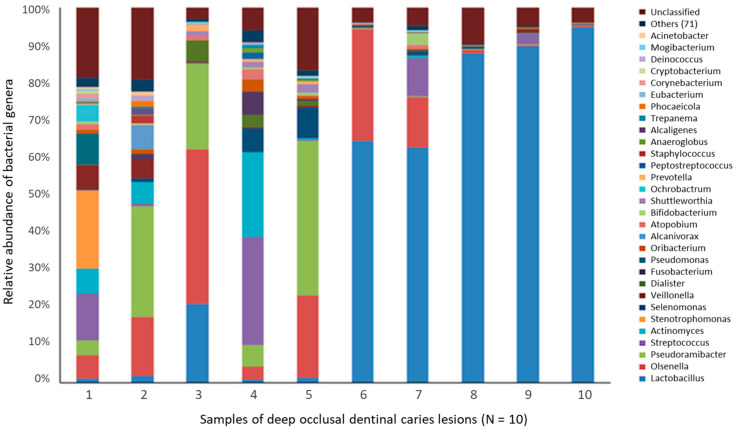
Relative abundance of the 30 most prevalent bacterial genera detected in advanced caries [191]. Bacterial samples were taken from deep occlusal (dentin) caries in permanent molars from 10 individuals. DNA was extracted for 16S rRNA gene (V4 variable region) sequence analysis. Approximately 347,646 partial 16S rRNA gene sequences were obtained. Overall, the *Lactobacillus* genus accounted for 42.3% of the sequences. The relative abundance per case in five samples ranged from 63% to 96% of the bacterial sequences.

**Figure 6 microorganisms-12-00121-f006:**
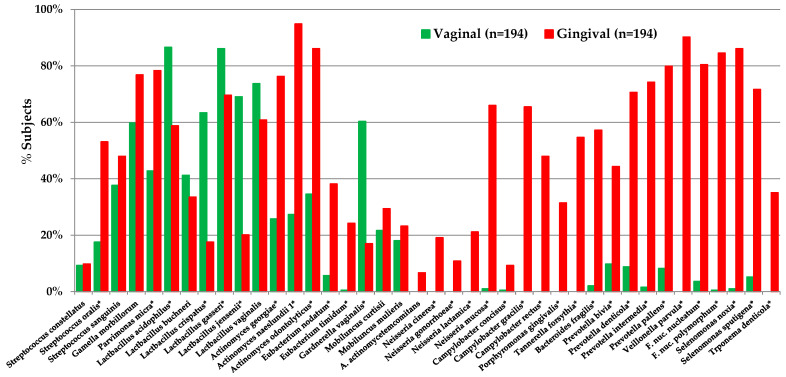
*Lactobacillus* species detected in vaginal and gingival samples. Oral (gingival) and vaginal samples were taken from 194 pregnant women in the first trimester [196]. Samples were analyzed using DNA gene probes in a checkerboard format with a 10^5^ threshold of species detection. Of the *Lactobacillus* species assayed, most were detected in both sample sites, suggesting that the vagina could be a source of oral lactobacilli in infants. In contrast, other species typical of subgingival sites, including *Porphyromonas* and *Prevotella* species, were detected more frequently in the gingival samples. * Difference detection, Chi-square > 0.05.

**Figure 7 microorganisms-12-00121-f007:**
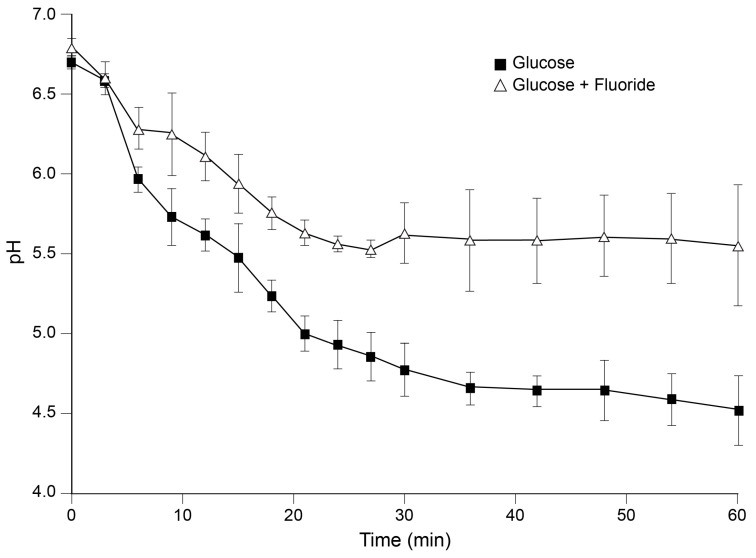
Acid production from a glucose solution with added fluoride on in vitro dental biofilms. Effects of a glucose solution with added fluoride on the in vitro pH of dental biofilms. pH dropped in 10-day biofilms after an overlay of glucose or glucose + 10 ppm fluoride [247].

**Figure 8 microorganisms-12-00121-f008:**
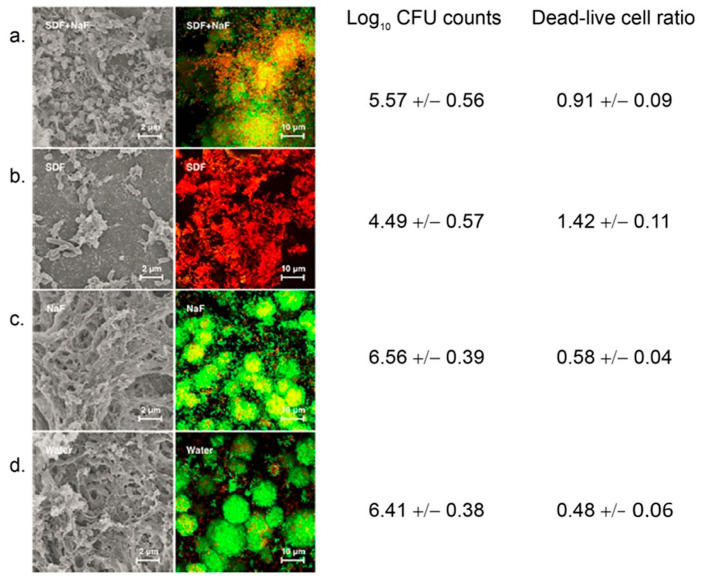
Effects of sodium diamine fluoride (SDF) on *S. mutans* biofilms cultured on dentin blocks in vitro. Shown are scanning electron micrographs (SEM, left panels) of the biofilm topography and the images from confocal laser scanning microscopy (CSLM, right panels) of the *S. mutans* biofilm on dentin caries lesions treated with (**a**) 38% SDF solution plus 5% NaF varnish; (**b**) 38% SDF solution alone; (**c**) 5% NaF; and (**d**) a water-treatment control. The green color represents live bacterial cells; the red color represents dead bacterial cells (magnification ×1000). The green intensity represents the amount of biofilm formed in the carious lesions. Modified based on Yu et al. 2018 [279].

## Data Availability

Data are from literature sources as cited in the manuscript.

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
