# Peer review of "The Evolving Microbiome of Dental Caries"

_microorganisms, 2024, doi:10.3390/microorganisms12010121_

Round 1

Reviewer 1 Report

Comments and Suggestions for Authors

This manuscript comprehensively reviewed the epidemiology of dental caries and the oral microbiota involved in different stage of caries development, as well as newly discovered treatment strategies for modulating microbial dysbiosis in dental caries. This review is well written. Please see some minor comments below:

1. In page 6, line 250: "CV Black" should be "GV Black".

2. In page 9, the first paragraph (line 391-405) is repetitive with the last paragraph in page 8 (line 363-376).

Author Response

Thank you for your comments, we have revised it accordingly.

Reviewer 2 Report

Comments and Suggestions for Authors

A literature review of the major pathologies of the oral cavity, well structured, updated, specific and detailed.

In my opinion, as a doctor, it provides a very important repository of information.
Maybe a little long. The type of review and the characteristics of the research should be identified. Narrate. However, we must set standards.

I have no changes to make to the scientific content. I just think that authors should include criteria even for the type of review presented here.

Author Response

Reviewer 1

  1. A literature review of the major pathologies of the oral cavity, well structured, updated, specific and

In my opinion, as a doctor, it provides a very important repository of information. Maybe a little long. The type of review and the characteristics of the research should be identified. Narrate. However, we must set standards.

Response: the text has been shortened by consolidating and rearranging some text as suggested by Reviewer #3 (see response to Reviewer #3 below).

  1. I have no changes to make to the scientific content. I just think that authors should include criteria even for the type of review presented

Response: Lines 74-75. The introduction has been edited to indicate that “the review is based on literature searches under the categories of caries microbiology, epidemiology, etiology, and

treatment.”

Reviewer 3 Report

Comments and Suggestions for Authors

Dear Authors,

This paper addresses an interesting topic. It is not easy to make revision of this comprehensive review. 

Only minor changes are required before publication:

Figure 1. I suggest improving readability by enlarging each graph. 

Figure 2. Abbreviation “MMP” should be defined.

Figure 3. Vertical orientation should improve readability. 

Great job!

Best regards and good luck

Author Response

Reviewer 2

This paper addresses an interesting topic. It is not easy to make revision of this comprehensive review. Only minor changes are required before publication.

  1. Figure I suggest improving readability by enlarging each graph.

Response: Figure 1 has been revised by increasing the text size to improve readability.

  1. Figure Abbreviation “MMP” should be defined.

Response: The Figure 2 legend has been revised and no longer includes the acronym “MMP”

  1. Figure Vertical orientation should improve readability.

Response: Figure 3 has been revised by changing the orientation and including only identifications to the species but not the genus level.

Reviewer 4 Report

Comments and Suggestions for Authors

I revised the manuscript “The Evolving Microbiome of Dental Caries”, a narrative review that aims to illustrate the evolution of knowledge of the oral microbiome associated with dental caries, from the earliest to the present. The manuscript also provides a present and past epidemiological overview and illustrates current therapeutic/preventive options.

The manuscript is well written, well-structured and clear. I congratulate the Authors on a job well done.

Below are just a few small suggestions:

Abstract: I don't really agree with the sentence “Due to its multifactorial nature, caries has been difficult to treat successfully”. In clinical practice, we are able to treat caries successfully. Perhaps the authors wanted to refer to prevention and not to treatment. This possible mismatch of terms treatment/prevention I also found in other sentences in the section 'treatment of caries as a microbial disease'. I therefore invite the Authors to check the use of the two terms as they are not synonymous.

Introduction: The opening part of the introduction is totally devoid of any reference. Add references to support what is written (lines 31-42).

Background:

-        I would suggest numbering the sub-sections (e.g.: 2.1, 2.2, 2.3, etc.) throughout the manuscript in order to help the reader interested in certain sections specifically to interrupt and possibly resume reading later.

-        Add the 2015-2016 Survey reference (line 76).

-        The images in Figure 1 are very small and it is difficult to read the x-axis.

-        I would suggest moving, for ease of reading, lines 89-117 to the beginning of paragraph 2.1 or in the introduction as they are discursive concepts mentioned earlier and less suitable to be placed after a series of epidemiological data.

Microbiome of dental caries:

-        Recent scientific research has shown that not only the microbiome of the GI tract, but also the oral microbiome, affects general health as well as oral health, again contributing to the development of OSCC as reported in a recent systematic review.

Di Spirito F, Di Palo MP, Folliero V, Cannatà D, Franci G, Martina S, Amato M. Oral Bacteria, Virus and Fungi in Saliva and Tissue Samples from Adult Subjects with Oral Squamous Cell Carcinoma: An Umbrella Review. Cancers (Basel). 2023 Nov 22;15(23):5540. doi: 10.3390/cancers15235540.

I would suggest specifying this aspect (oral microbiome-health-OSCC) as well, as done for the GI microbiome, the article being focused on the oral microbiome.

-        I would suggest removing and replacing the word 'dramatically' (line 244) with another word that has a positive meaning (also in the conclusion).

-        Write the abbreviation NIH in full on line 282 where it appears for the first time.

-        The caption of figure 2 should only contain the information necessary for the reader to understand what is depicted in the figure. I suggest moving much of the caption into the text, as much of what is written is not descriptive of the figure.

-        Lines 392-402 are the same as lines 363-376.

-        Line 434: replace the comma with a full stop after the reference.

-        The word “Sjögren” is badly written. Correct it.

-        Figure 3: Explain what the y-assis is.

-        Standardize the acronym for “severe childhood caries” à SECC or S-ECC.

-        Write the abbreviation PCR (line 905), and MALDI-TOP.

References: The references are not formatted in accordance to the guidelines journal.

I would also like to point out to the authors that two names are listed in the 'correspondence' as corresponding author, but the asterisk is only next to one name.

Author Response

Reviewer 3

In the manuscript “The Evolving Microbiome of Dental Caries”, a narrative review that aims to illustrate the evolution of knowledge of the oral microbiome associated with dental caries, from the earliest to the present. The manuscript also provides a present and past epidemiological overview and illustrates current therapeutic/preventive options.

The manuscript is well written, well-structured and clear. I congratulate the Authors on a job well done. Below are just a few small suggestions:

Abstract:

  1. Line 23: I don't really agree with the sentence “Due to its multifactorial nature, caries has been difficult to treat successfully”. In clinical practice, we are able to treat caries successfully. Perhaps the authors wanted to refer to prevention and not to

Response: line 23 The word “prevent” has been replaced with “treat successfully” in the abstract.

  1. There’s a possible mismatch of terms treatment/prevention which I also found in other sentences in the section 'treatment of caries as a microbial disease'. I therefore invite the Authors to check the use of the two terms as they are not synonymous.

Response:

Line 1312: the term “intervention” has replaced “treatment”.

Line 1341: “treatment” has been deleted and replaced with “application of” before SDF Line 1397: “treatment” has been replaced with “treatment and prevention approaches” Line 1503 and 1526: “treatment” has been deleted leaving regimen/regimens

Introduction:

  1. The opening part of the introduction is totally devoid of any reference. Add references to support what is written (lines 31-42).

Response: the introduction has been revised as suggested by the reviewer (see below), and the section without references has since been deleted.

Background:

  1. I would suggest numbering the sub-sections (e.g.: 2.1, 2.2, 2.3, etc.) throughout the manuscript in order to help the reader interested in certain sections specifically to interrupt and possibly resume reading

Response: sub-section numbers have been added following the “instructions to authors”.

  1. Add the 2015-2016 Survey reference (line 76).

Response: (new line 104) The reference has been added for the 2015-2016 survey data.

  1. The images in Figure 1 are very small and it is difficult to read the X-axis.

Response: Figure 1 has been revised as suggested by Reviewer #2 above.

  1. I would suggest moving, for ease of reading, lines 89-117 to the beginning of paragraph 2.1 or in the introduction as they are discursive concepts mentioned earlier and less suitable to be placed after a series of epidemiological data.

Response: Sections from lines 89-117 have been moved to the introduction, with editing to avoid duplication of text. This resulted in line number changes.

Microbiome of dental caries:

  1. Recent scientific research has shown that not only the microbiome of the GI tract, but also the oral microbiome, affects general health as well as oral health, again contributing to the development of OSCC as reported in a recent systematic

Di Spirito F, Di Palo MP, Folliero V, Cannatà D, Franci G, Martina S, Amato M. Oral Bacteria, Virus and Fungi in Saliva and Tissue Samples from Adult Subjects with Oral Squamous Cell Carcinoma: An Umbrella Review. Cancers (Basel). 2023 Nov 22;15(23):5540. doi: 10.3390/ca.ncers15235540.

I would suggest specifying this aspect (oral microbiome-health-OSCC) as well, as done for the GI microbiome, the article being focused on the oral microbiome.

Response: The reviewer makes an excellent point about associations between the oral microbiota and cancers. Although the authors considered this concept for inclusion in the text, ultimately, they decided that the Di Spirito et al reference did not fit well into this review since it is more likely that

the periodontal microbiota – and less so the caries microbiota - that is cancer associated. The caries microbiota is the focus of this review.

  1. I would suggest removing and replacing the word 'dramatically' (line 244) with another word that has a positive meaning (also in the conclusion).

Response: Line 241 (original line 244), the term “dramatically” was changed to “markedly”.

Line 1553 in Conclusions “advanced dramatically” has been changed to “greatly advanced".

  1. Write the abbreviation NIH in full on line 282 where it appears for the first

Response: (Line 280) NIH has now been defined where it is mentioned for the first time.

  1. The caption of figure 2 should only contain the information necessary for the reader to understand what is depicted in the I suggest moving much of the caption into the text, as much of what is written is not descriptive of the figure.

Response: Figure 2 legend has been amended to reflect only what is depicted in the figure as suggested proposed by the reviewer. Text that did not directly relate to the figure was deleted from the legend because it was either already mentioned in the text, or did not contribute significantly to the microbial succession of caries.

  1. Lines 392-402 are the same as lines 363-376.

Response. We thank the reviewers for noticing this. The duplicated text has been deleted.

  1. Line 434: replace the comma with a full stop after the

Response: The comma after the reference in what is now line 431 has been replaced by a full stop/period.

  1. The word “Sjögren” is badly

Response. This correction has been made and it appears in line 626 of the revised manuscript.

  1. Figure 3: Explain what the y-axis

Response: The X and Y axes are now appropriately labelled in Figure 3.

  1. Standardize the acronym for “severe childhood caries” à SECC or S-ECC.

Response: The text for severe childhood caries has been standardized to S-ECC. We thank this reviewer for noticing this.

  1. Write out the abbreviations for PCR (line 905), and MALDI-TOF.

Response: PCR (Line 916) and MALDI-TOF (line 918) have been defined in the revision.

  1. References: The references are not formatted in accordance to the guidelines

Response: The correct reference style has been applied in the revision.

  1. I would also like to point out to the authors that two names are listed in the 'correspondence' as corresponding author, but the asterisk is only next to one

Response: This has been corrected. An asterisk now appears for both corresponding authors.

Round 2

Reviewer 4 Report

Comments and Suggestions for Authors

Good job and good luck